# Identification of Latent Confounders via Investigating the Tensor Ranks of the Nonlinear Observations

**Zhengming Chen** [1 2]  **Yewei Xia** [3 4]  **Feng Xie** [5]  **Jie Qiao** [2]  **Zhifeng Hao** [1]  **Ruichu Cai** [2]  **Kun Zhang** [4 6]

## Abstract

We study the problem of learning discrete latent variable causal structures from mixed-type observational data. Traditional methods, such as those based on the tensor rank condition, are designed to identify discrete latent structure models and provide robust identification bounds for discrete causal models. However, when observed variables—specifically, those representing the children of latent variables—are collected at various levels with continuous data types, the tensor rank condition is not applicable, limiting further causal structure learning for latent variables. In this paper, we consider a more general case where observed variables can be either continuous or discrete, and further allow for scenarios where multiple latent parents cause the same set of observed variables. We show that, under the completeness condition, it is possible to discretize the data in a way that satisfies the full-rank assumption required by the tensor rank condition. This enables the identifiability of discrete latent structure models within mixed-type observational data. Moreover, we introduce the two-sufficient measurement condition, a more general structural assumption under which the tensor rank condition holds and the underlying latent causal structure is identifiable by a proposed two-stage identification algorithm. Extensive experiments on both simulated and real-world data validate the effectiveness of our method.

---

[1]College of Mathematics and Computer, Shantou University, Shantou, China [2]School of Computer Science, Guangdong University of Technology, Guangzhou, China [3]Department of Computer Science, Fudan University, Shanghai, China [4]Department of Machine Learning, Mohamed Bin Zayed University of Artificial Intelligence, Abu Dhabi, UAE [5]Department of Applied Statistics, Beijing Technology and Business University, Beijing, China [6]Department of Philosophy, Carnegie Mellon University, Pittsburgh, USA. Correspondence to: Ruichu Cai <cairuichu@gmail.com>, Kun Zhang <kunz1@cmu.edu>.

*Proceedings of the $42^{nd}$ International Conference on Machine Learning*, Vancouver, Canada. PMLR 267, 2025. Copyright 2025 by the author(s).

## 1. Introduction

Latent confounders, typically referred to as unobserved variables that influence multiple observed variables, are frequently encountered in various disciplines such as statistics, machine learning, and causal discovery. Ignoring latent confounders may introduce spurious correlations among observed variables, impeding the correctness and robustness of models trained from observational data. Taking into account latent confounders in modeling procedures remains a long-standing challenge and has attracted wide attention in both theoretical and applied research (Zheng et al., 2023; Li et al., 2024; Miao et al., 2018; Thaden & Kneib, 2018; Wang & Blei, 2019; Bartolucci et al., 2023).

Much effort has been made to handle the latent confounding problem in causal structure learning. One line of research considers the causal structure over the measured variables, including constraint-based methods (Spirtes et al., 2000; Colombo et al., 2012; Zhou et al., 2020), functional model-based ones (Chen et al., 2021b; Hoyer et al., 2008; Salehkaleybar et al., 2020), and hybrid model approaches (Chen et al., 2021c; Ogarrio et al., 2016). Another line of research focuses on identifying the causal structure among latent variables by utilizing the pure measurement model assumption, where each latent variable has certain measured variables as its children. This includes linear model-based methods (Silva et al., 2006; Cai et al., 2019; Xie et al., 2020; Chen et al., 2022; Jin et al., 2023; Huang et al., 2022; Xie et al., 2024) and discrete latent tree model-based approaches (Gu, 2022; Gu & Dunson, 2023; Choi et al., 2011; Zhou et al., 2020). These methods either assume linear relationships among all variables or constrain the latent structure to a specific structure. Consequently, they are unable to handle a general causal structure with non-linear relationships among latent variables and observed variables.

When observed variables are continuous and nonlinearly depend on the discrete latent variables, one straightforward way to learn the causal structure among latent variables is to treat it as a mixture model (Teicher, 1967; Tahmasebi et al., 2018; Allman et al., 2009; Ritchie et al., 2020), so that the causal structure can be learned from the recovered latent distribution (Kivva et al., 2021). However, such recovery requires an unbiased estimation of the parameters

of the mixture model, which is unrealistic and impractical (Chung et al., 2004; xian Wang et al., 2004). Recent work by (Chen et al., 2024) shows that it is possible to identify causal relations among latent variables by recovering the cardinality of the support of latent variables, which is rather practical and also testable. However, it is only applicable in discrete cases, and it is still unclear whether the results hold in the presence of continuous observed variables.

In this paper, we investigate the problem of identifying discrete latent variables and their causal structure from non-linearly dependent observed variables. When the observed variables are also discrete, the tensor rank condition (Chen et al., 2024) implies $d$-separation by identifying the cardinality of the latent supports from observed variables. This raises the question: can the cardinality of latent supports be identified from continuously observed variables, and then can the tensor rank condition be applied to further learn the causal structure? Interestingly, we find that it is possible to identify the cardinality of latent supports from nonlinearly dependent observed variables (whether discrete or continuous) by, for example, using appropriate discretization techniques. We show that, under the completeness condition, the tensor rank condition can be used to identify the $d$-separation among observed variables that are non-linearly dependent given the discrete latent confounder, if there are two sufficient measured variables for each latent variable. Based on this, we apply the tensor rank condition to the discrete latent structure model with mixed-type observational data and develop a two-stage identification algorithm to learn a more general discrete latent variable structure, where an observed variable can have multiple latent parents. We theoretically show that the latent structure of the discrete latent structure model is identified up to a Markov equivalent class by properly utilizing the tensor rank condition.

The contributions of this work are three-fold: (1) We first demonstrate that under the completeness condition, the tensor rank condition holds for variable sets that may be non-linearly dependent on latent confounders. (2) We develop an identification algorithm that identifies a more general discrete latent variable structure up to a Markov equivalent class. (3) We conduct simulated experiments to verify its capability to handle mixed-type observational data.

## 2. Non-linear Causal Models with Discrete Latent Confounders

In this paper, we focus on the causal graphical model (Spirtes et al., 2000) and study how to identify latent variables and their causal relationships from measured variables that may be nonlinearly dependent on latent variables. We first introduce the basic notation.

Consider a causal graph $\mathcal{G} = \{\mathbf{V}, \mathbf{E}\}$, it is a DAG with

$\mathbf{V} = (\mathbf{X}, \mathbf{L})$, where $\mathbf{X} \in \mathbb{R}^n$ is the observed variable set while $\mathbf{L} \in \Omega^m$ is the latent variable set, $\Omega$ is a discrete space with $|\Omega| = r \geq 2$. In a causal graph, we do not allow directions from observed to latent to preserve the latent confounder structure. The data generation of observed variables is defined as

$$X_i := f(Pa(X_i), \varepsilon_{X_i}), \tag{1}$$

where $Pa(X_i)$ is the parent set of $X_i$ and $\varepsilon_{X_i}$ is noise term of $X_i$. We denote the generating function by $f$ : $[Pa(X_i), \varepsilon_{X_i}] \mapsto X_i$. We denote such a causal model as the *nonlinear causal model with discrete latent confounders*.

In a causal model, certain causal assumptions are necessary to ensure identifiability. We assume that the distribution $\mathbb{P}(\mathbf{V})$ satisfies the Markov property and Faithfulness condition with respect to $\mathcal{G}$. That is, the distribution $\mathbb{P}(\mathbf{V}) = \prod_{V_i \in \mathbf{V}} \mathbb{P}(V_i | Pa(V_i))$ and the conditional independence relations imply the $d$-separation structure in the causal graph (Spirtes et al., 2000). Besides, to identify the discrete latent variable only from observational data, we make the following non-degeneracy assumption.

**Assumption 2.1** (Non-degeneracy). The distribution over $\mathbf{V} = \{\mathbf{X}, \mathbf{L}\}$ satisfies:

(a) [Non-zero mass condition] For any $r \in \Omega$, $\mathbb{P}(L_i = r) > 0$ for all $L_i \in \mathbf{L}$.
(b) [Full-rank condition] For any discrete conditional distribution contingency table $\mathbb{P}(V_i | Pa(V_i))$ is full rank.

The non-degeneracy assumption has also been used in (Kivva et al., 2021; Kong et al., 2024; Chen et al., 2024) to ensure the identifiability of discrete latent variables. Further discussion on this topic can be found in (Kivva et al., 2021).

Throughout this paper, we use standard notation such as $Pa(V_i) = \{V_j | V_j \rightarrow V_i\}$, $Ch(V_i) = \{V_j | V_i \rightarrow V_j\}$, $Anc(V_i) = \{V_j | V_j \rightsquigarrow V_i\}$, $Des(V_i) = \{V_j | V_i \rightsquigarrow V_j\}$ to denote the set of parents, children, ancestors, descendants, and nodes of $V_i$, respectively. For a discrete variable $V_i$, we use $\mathrm{supp}(V_i) = \{v \in \mathbb{Z}^+ : \mathbb{P}(V_i = v) > 0\}$ to denote the set of possible values of $V_i$, and $\mathcal{T}_{(\mathbf{X}_p)}$ to denote probability tensor of discrete joint distribution $\mathbb{P}(\mathbf{X}_p)$. Beside, we use $|\mathbf{V}_p|$ denote the dimension (or cardinality) of $\mathbf{V}_p$.

**Goal.** We aim to develop a statistically testable method to learn the causal structure among latent variables in the nonlinear causal model with discrete latent confounders.

## 3. Graphical Criteria in Non-linear Causal Models with Discrete Latents

We begin with a brief review of discrete latent structure learning methods, with a focus on the tensor rank condition. Subsequently, we demonstrate that the tensor rank condition

can be extended to nonlinear causal models with discrete latent confounders. Thus, we provide a general result for the tensor rank condition (Theorem 3.4).

## 3.1. Background

When the mixture model over $\mathbf{X}$ is identifiable, (Kivva et al., 2021) shows that the causal structure among latent variables is identifiable from an existing mixture oracle. To estimate the parameters of the mixture oracle, such as the number of components, (Kivva et al., 2021) applies a K-means algorithm. However, since K-means is an approximate method, it may sometimes misidentify the number of components in real-world applications, potentially leading to incorrect identification of the causal structure.

*Example* 3.1 (Incorrect identifying the cardinality of latent support). As shown in Fig. 1, the left side represents the discrete latent DAG, while the right side shows the corresponding mixture distribution. In the latent structure, the latent variables $L_1, L_2$ have the same support $\{0, 1\}$, and the observed variable is mixed by the Laplace distribution. One can see that the cardinality of the latent variable is incorrectly identified as three (when it is actually four) by the K-means algorithm (Kivva et al., 2021), even in a simple structure (left side in Fig .1).

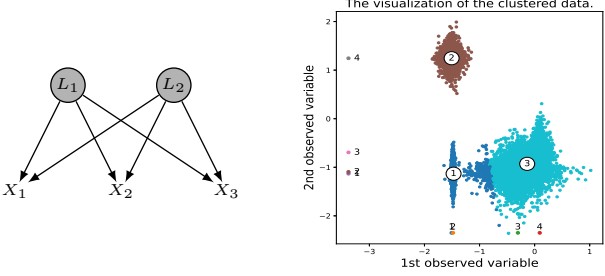

*Figure 1.* Example of a discrete latent DAG and corresponding mixture distribution, clustered by K-means algorithms.

While the identification of latent variables can be built upon the identifiability of mixture models, learning mixture models is a nontrivial problem (Kivva et al., 2021). Rather than relying on approximate methods to estimate mixture models, seeking a simple and robust approach is more meaningful. Recently, (Chen et al., 2024) introduced a statistically testable tool, the tensor rank condition, for handling discrete cases with discrete latent confounders. By examining the rank of the probability tensor, which reflects the cardinality of the latent support, the tensor rank condition establishes a connection between algebraic constraints and the $d$-separation structure in discrete latent structure models, and can be used to identify the latent structure. Mathematically, the graphical implication of the tensor rank condition is as follows (Theorem 3.2).

**Theorem 3.2** (Graphical implication of tensor rank condition (Chen et al., 2024)). *In the discrete causal model, suppose the Markov, Faithfulness condition and full-rank condition hold. Consider an observed variable set $\mathbf{X}_p = \{X_1, \cdots, X_p\}$ ($\mathbf{X}_p \subseteq \mathbf{X}$ and $2 \leq p \leq n$) and the corresponding $p$-way probability tensor $\mathcal{T}_{(\mathbf{X}_p)}$ that is the tabular representation of the joint probability mass function $\mathbb{P}(X_1, \cdots, X_p)$, then $\mathrm{Rank}(\mathcal{T}_{(\mathbf{X}_p)}) = r$ ($r > 1$)[1] if and only if (i) there exists a variable set $\mathbf{S} \subset \mathbf{V}$ with $|\mathrm{supp}(\mathbf{S})| = r$ that $d$-separates any pair of variables in $\{X_1, \cdots, X_p\}$, and (ii) does no exist conditional set $\tilde{\mathbf{S}}$ that satisfies $|\mathrm{supp}(\tilde{\mathbf{S}})| < r$.*

The *full-rank assumption* (in Theorem 3.2) posits that any conditional probability table $\mathbb{P}(V|\mathrm{Pa}(V))$ is full rank, aligned with the non-degeneracy condition (b). Roughly speaking, the tensor rank condition shows that one can identify the cardinality of latent support by the probability tensor of observed variables, for detecting the $d$-separation relations among observed or latent variables. However, when the observed variables are continuous, which exhibit more general nonlinear dependencies, it remains unclear under what conditions the graphical implications of the tensor rank condition hold for the probability tensor over continuous variables. This is because the graphical criteria rely on the full-rank assumption, which typically only applies to discrete variables. Fortunately, we find a well-studied condition, the *completeness condition*, under which the graphical implications of the tensor rank condition also hold for the continuous observed variables.

*Condition* 3.3 (Completeness condition). For any $X_i \in \mathbf{X}$ that has only one (latent) parent $L_j$, we assume that the conditional distribution $\mathbb{P}(L_j|X_i)$ is complete. That is, for all measurable real functions $g$ such that $\mathbb{E}(|g(l)|) < +\infty$, $\mathbb{E}(g(l)|x) = 0$ almost surely iif. $g(l) = 0$ almost surely.

In this case, any function $g : \Omega \to R$ is indeed bounded. This condition is also used in (Miao et al., 2018; Cui et al., 2023; Liu et al., 2024). It is worth noting that many commonly-used parametric and semiparametric models such as exponential families (Newey & Powell, 2003) and location-scale families like convolution density functions (Hu & Shiu, 2018) satisfy the completeness condition. Further discussion can be found in Appendix.

## 3.2. General Results for Tensor Rank Condition

Here, we aim to show that the tensor rank condition holds in the nonlinear causal model with discrete latent confounders under the completeness condition.

---

[1] The rank of a tensor $\mathcal{T}$, denoted $\mathrm{Rank}(\mathcal{T})$, is the smallest number of rank-one tensors that generate as their sum, where a $N$-way tensor is rank-one if it can be written as the outer product of $N$ vectors.

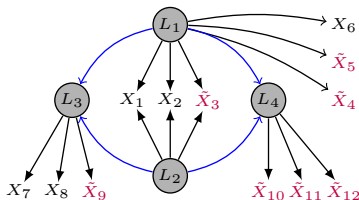

*Figure 2.* Example of discrete latent structure model with mixed-type observational data, where $X_i$ represents continuous observed variables $\tilde{X}_i$ represents discrete observed variables (purple), and $L_i$ represents a discrete latent variable. Moreover, the blue edge constructs the structure model $\mathcal{C}$, the black edge constructs the measurement model $\mathcal{M}$.

**Theorem 3.4** (Graphical Implication in Non-linear Causal Model). *In the nonlinear causal model with discrete latent confounders, suppose the Markov condition, faithfulness, non-degeneracy condition, and the completeness condition hold. Consider an observed variable set $\mathbf{X}_p = \{X_1, \ldots, X_p\}$, $\mathbf{X}_q \subset \mathbf{X}$ is a continuous observed variable set (can be empty), for its corresponding $p$-way (conditional) probability tensor $\mathbb{P}(\mathbf{X}_p|\mathbf{X}_q)$, denoted by $\mathcal{T}_{(\mathbf{X}_p|\mathbf{X}_q)}$, then $\mathrm{Rank}(\mathcal{T}_{(\mathbf{X}_p|\mathbf{X}_q)}) = r$ and $r \ll p$, if and only if (i) there exists a variable set $\tilde{\mathbf{S}} \subset \mathbf{V}$ with $|\mathrm{supp}(\tilde{\mathbf{S}})| = r$ such that $\tilde{\mathbf{S}} \cup \mathbf{X}_q = \mathbf{S}$, $\mathbf{S}$ $d$-separates any pair variables in $\mathbf{X}_p$ (conditional on $\mathbf{X}_q$ if $\mathbf{X}_q \neq \emptyset$), and (ii) does no exist conditional set $\tilde{\mathbf{S}} \cup \mathbf{X}_q$ that satisfies $|\mathrm{supp}(\tilde{\mathbf{S}})| < r$.*

The above theorem shows that in the nonlinear causal model with discrete latent confounders, given an observed variable set $\mathbf{X}_p$, the rank of its (conditional) probability tensor corresponds to the cardinality of the discrete conditional variable set that $d$-separates all variables in $\mathbf{X}_p$. When all observed variables are discrete, Theorem 3.2 is a specific case of Theorem 3.4. Moreover, we allow continuous, nonlinearly dependent observed variable sets and further consider the constraints on the (conditional) probability tensor.

*Remark* 3.5. Although computing the tensor rank for continuous, nonlinearly dependent observed variables is intractable due to the curse of dimensionality, we will show in Section 4 that a practical implementation is to discretize the continuous variables, thereby making tensor rank computation feasible.

# 4. Application in Discrete Latent Structure Models

In this section, our goal is to apply the tensor rank condition (Theorem 3.4) to perform causal discovery with latent variables in the nonlinear causal model with discrete latent confounders. Without additional assumptions, however, it is hard to identify the causal structure among latent variable $\mathbf{L}$ only from observed variables $\mathbf{X}$ (Silva et al., 2006; Kivva et al., 2021). Generally speaking, to address this issue, one

common model is to adopt the pure measurement model, such as the linear case (Silva et al., 2006; Cai et al., 2019; Xie et al., 2020) and discrete case (Chen et al., 2024; Gu, 2022).

Based on the nonlinear causal model with discrete latent confounders and combining the discrete latent structure model from (Chen et al., 2024), in this paper, we further consider a more challenging case where the observed variable can be discrete or continuous, and allow for a multi-factor structure among observed variables. We formalize this as the discrete latent structure model with mixed-type data.

**Definition 4.1** (Discrete Latent Structure Model with Mixed–Type Observational Data). A nonlinear causal model with latent confounders and its corresponding causal graph $\mathcal{G}$ is a discrete latent structure model with mixed-type data (Mixed LSM) if it satisfies the following conditions:

- (Purity Assumption) there are no direct edges between the observed variables;
- (Three-Pure Child Variable Assumption) each latent variable set $\mathbf{L}_p \subset \mathbf{L}$ (at least exist one $|\mathbf{L}_p| = 1$), in which every latent variable directly causes the same set of observed variables, has at least $2|\mathbf{L}_p| + 1$ pure variables[2] as children;
- (Two-Sufficient Measurement Assumption) each latent variable set $\mathbf{L}_p \subset \mathbf{L}$, in which every latent variable directly causes the same set of observed variables, has at least two sufficient measured variables[3], which can be either continuous variables or discrete variables with larger support than the latent parents.

Note that the mixed LSM is different from the discrete latent variable model like (Chen et al., 2024; Gu, 2022), in which they assume all variables are discrete variables. The mixed-type setting more closely aligns with real-world scenarios, where the latent variable of interest is assessed using multiple indicators at varying measurement levels. For instance, a specific disease (unobserved, discrete variable) may be reflected by various physiological indicators, such as blood oxygen levels (continuous) and heart rate (discrete).

**Discussions on the Assumptions.** In general, to study the causal structure of latent variables, the purity and three-pure-children assumption are commonly used, even in linear models (Silva et al., 2006; Kummerfeld & Ramsey, 2016; Chen et al., 2022). When the latent variable is discrete, some works achieve identification under weaker assumptions, such as the two-pure-children assumption (Gu, 2022) and the multi-parent assumption (Kivva et al., 2021; Kong et al., 2024). However, these approaches often involve spe-

---

[2]Pure variables denote the variables that have only one latent parent, and no observed parents.

[3]For an observed variable $X$ with support $\mathcal{X}$ and latent parent $L$ with support $\Omega$, we define a sufficient measurement as $|\mathcal{X}| > |\Omega|$.

cific structural constraints or complex estimation algorithms, hindering their practical application in real-world scenarios. It is important to emphasize that the purity and three-pure-children assumption are only sufficient conditions for identifying the causal structure of latent variables, which ensures the identification algorithm is simpler and more efficient. In fact, relaxing these assumptions to a more general case is feasible. Further discussion on this extension is provided in Appendix D.

Under the mixed-type LSM, $\mathcal{G}$ decomposes as the union of two subgraphs $\mathcal{G} = \mathcal{M} \cup \mathcal{S}$, where $\mathcal{M}$ denotes the measurement model graph that represents the bipartite graph of edges pointing from $\mathbf{L}$ to $\mathbf{X}$, and $\mathcal{S}$ represents the structure model graph that is a DAG over the latent variable $\mathbf{L}$. Such a definition can be referred to previous works (Chen et al., 2024; Silva et al., 2006; Cai et al., 2019; Xie et al., 2020; Gu, 2022). See Figure 2 for example.

Based on the graphical implication of the tensor rank condition, one can develop a statistically testable and robust algorithm to recover the latent structure, aligning with (Chen et al., 2024). However, the challenge hindering the application of the tensor rank is that for the probability tensor involving continuous variables, it is intractable to compute the tensor rank due to the curse of dimensionality. Thus, we first provide a practical method to test the tensor rank condition for the continuous probability tensor by employing proper discretization techniques.

### 4.1. Estimation Tensor Rank from Continuous Observed Variables

To apply the tensor rank condition (Theorem 3.4) for continuous observed variables, one practical way is to properly discretize continuous variables and then test the tensor rank condition on the discretized data. However, to ensure the tensor rank condition holds in the discretized data, two key issues must be addressed:

1). whether there exists a discretization of continuous data such that the tensor rank condition holds, and
2). how to implement such discretization.

Resolving the first issue guarantees that a suitable discretization approach can be identified to apply the tensor rank condition for causal discovery, while the second issue focuses on the operable implementation procedure. We begin by addressing the first issue.

For the discrete data, the tensor rank condition relies on the full-rank condition, as shown in Theorem 3.2. Therefore, to apply the tensor rank condition to discretized data (derived from continuous observed variables), the task is equivalent to finding a discretization such that $\mathbb{P}(\tilde{X}_i | Pa_{X_i})$ is full rank in the mixed LSM. We first show its existence.

**Proposition 4.2.** *In the mixed LSM, suppose that the Markov condition, faithfulness assumption, and completeness condition hold. Then, there exists a discretization $\tilde{X}_i$ of $X_i$, such that $\mathbb{P}(\tilde{X}_i | Pa_{X_i})$ has full rank.*

*Proof of Sketch.* In the mixed LSM, each observed variable is caused by their latent parents, i.e., $\mathbb{P}(\tilde{X}_i | Pa_{X_i}) = \mathbb{P}(\tilde{X}_i | L_i)$. Under the completeness condition, $\mathbb{E}(g(l)|x_i) = 0$ with $g(l) = 0$ implies that the linear independence of functions $\{f_{L_i|X_i}(l|x_i)\}_{l=1}^r$, where $f_{L_i|X_i}$ can be seen as the PMF of conditional distribution $\mathbb{P}(L_i|X_i)$. Therefore, one can further infer that $\{F_{X_i|L_i}(x_i|l)\}_{l=1}^r$ are also linear independent based on the Bayes theorem, where $F_{X_i|L_i}$ is the CDF of $\mathbb{P}(X_i|L_i)$. It ensures that the partition of $\mathbb{P}(X_i|L_i)$ is also linearly independent.

Intuitively, the completeness condition admits linear independence within the conditional probability space and maintains this independence for any subspace of the probability space. This induces the possibility to find a discretization that satisfies the full-rank assumption.

Now, we aim to address the second issue: how to implement such a discretization. However, it is a non-trivial task to find such a discretization that satisfies the full-rank condition in the mixed LSM. The challenge lies in the fact that $L$ of $\mathbb{P}(X|L)$ is a latent variable, which cannot be directly measured or tested.

*Remark* 4.3 (The challenging of discretization). Since one can only discretize the observed data $\mathbf{X}$ and $\mathbb{P}(X_i|L_i)$ is unobserved, it is possible to obtain discretized data in which $\mathbb{P}(\tilde{X}_i|L_i)$ is not full rank. When the full-rank condition does not hold in Theorem 3.4 (or, Theorem 3.2), the rank of the probability tensor may fail to uniquely represent the $d$-separation relations among the observed variables. Therefore, if continuous data is randomly discretized without careful consideration, the causal structure may be incorrectly identified using the tensor rank condition (see detailed example in Appendix).

Although the full-rank condition cannot be directly tested since $\mathbb{P}(X_i|L_i)$ cannot be obtained from observational data, we have found that it can be indirectly assessed by examining the non-negative rank of two discrete variables. This approach is motivated by the following observation.

**Lemma 4.4** (Upper bound after discretization)**.** *In the mixed LSM and suppose Markov condition, faithfulness assumption and full-rank condition hold. Let $\tilde{X}_i$ and $\tilde{X}_j$ be the discretization of $X_i$ and $X_j$, then we have $\mathrm{Rank}(\mathbb{P}(\tilde{X}_i, \tilde{X}_j)) \leq |\mathrm{supp}(L)|$, where $L$ is a latent parent that $d$-separates $X_i$ from $X_j$. Moreover, $\tilde{X}_i$ and $\tilde{X}_j$ satisfy the full-rank assumption if and only if the above inequality holds with equality.*

Lemma 4.4 shows that one can achieve the upper bound of the rank of probability tensor when the discretized variables

satisfy the full-rank condition. This motivates us to discretize continuous variables and then check their rank constraints over the joint contingency table, for achieving the full-rank condition. When the cardinality of latent support is unknown, one practical way is to use a rank-stop-increasing technique, which can indirectly detect the full-rank condition.

**Theorem 4.5** (Ranks' stopped increasing to implement full rank). *For a continuous random variable $X_i$ and a discrete variable $\tilde{X}_j$, denote $\tilde{X}_i$ be the discretization of $X_i$. Let $\mathbb{P}^{(k)}(\tilde{X}_i, \tilde{X}_j)$ be the k-th discretization of contingency tables $\mathbb{P}(\tilde{X}_i, \tilde{X}_j)$ that satisfies*

*1)* $\mathrm{Rank}(\mathbb{P}^{(k)}(\tilde{X}_i, \tilde{X}_j)) \geq \mathrm{Rank}(\mathbb{P}^{(k-1)}(\tilde{X}_i, \tilde{X}_j))$,
*2)* $\mathrm{Max}(\mathrm{Rank}(\mathbb{P}^{(k)}(\tilde{X}_i, \tilde{X}_j)) < \mathrm{Min}(|\mathrm{supp}(\tilde{X}_i)|, |\mathrm{supp}(\tilde{X}_j)|)$, *then there must exist a finite order $k$ such that*

$$\mathrm{Rank}(\mathbb{P}^{(k)}(\tilde{X}_i, \tilde{X}_j)) = \mathrm{Rank}(\mathbb{P}^{(k+1)}(\tilde{X}_i, \tilde{X}_j)). \quad (2)$$

*That is, $\tilde{X}_i$ and $\tilde{X}_j$ satisfy the full-rank assumption.*

*Remark* 4.6 (Practical implementations). We can discretize the continuous variable by choosing suitable cut-points (bin numbers) in practice. Generally speaking, a larger bin number effectively controls discretization error and more easily meets the full-rank assumption. To achieve this, we follow the heuristics proposed by (Dougherty et al., 1995) to set the cut points. Moreover, we can construct a hypothesis test for the null-hypothesis that $\mathbb{P}(\tilde{X}_i, \tilde{X}_j)$ has the rank $\geq r$, following the approach of (Mazaheri et al., 2023). This rank test method is based on matrix perturbation theory. More details can be found in (Ratsimalahelo, 2001).

In other words, one can discretize the data multi-times (sufficiently large $k$), to find the discretized data that satisfies the full-rank condition. Consequently, the tensor rank condition can then be applied to the discrete data to learn causal structure. Moreover, we can show that the conditional independence (CI) relations in the discretized data imply the true corresponding $d$-separation relations in the causal graph.

**Proposition 4.7.** *Let $\{\tilde{X}_1, \cdots, \tilde{X}_n\}$ be the discretized variable of $\{X_1, \cdots, X_n\}$ correspondingly and satisfy the full-rank condition. Then under the Markov assumption, faithfulness assumption, the CI relations among $\{\tilde{X}_1, \cdots, \tilde{X}_n\}$ imply the true $d$-separation in the causal graph $\mathcal{G}$.*

Based on Proposition 4.7, we can derive that the tensor rank condition and its graphical implications hold in the discretized data (i.e., data obtained from the discretization of continuous variables).

**Lemma 4.8** (Tensor rank condition in mixted-type data). *In the nonlinear causal model with latent confounders, suppose the Markov condition, faithfulness assumption, and non-degeneracy assumption hold, then for the continuous variable set $\mathbf{X}_p$ that satisfies the completeness condition,* *there exists a discretized dataset $\tilde{\mathbf{X}}_p$ of $\mathbf{X}_p$, such that the graphical implication of tensor rank condition holds.*

## 4.2. Structure Learning in Mixed LSMs

In this section, we aim to learn the causal structure of mixed latent structure models (LSMs) from mixed-type observational data. One commonly used approach is the two-stage structure learning algorithm, as seen in (Silva et al., 2006; Chen et al., 2024). Specifically, we first learn the causal clusters among the observed variables to determine the latent variables (Step I), and then test the $d$-separation relations among the latent variables using their measured variables to recover the causal structure of the latent variables (Step II).

### 4.2.1. STEP I: FINDING CAUSAL CLUSTERS

Without any constraints, the discrete latent variable is unidentifiable. When the observed variable set shares a common latent parent, for instance, one can always replace a pair of distinct latent variables $L_i$ and $L_j$ with a single latent variable $L_k$, where $|\mathrm{supp}(L_k)| = |\mathrm{supp}(L_i, L_j)|$. To simplify the identification algorithm, we assume all latent variables have the same support and that at least one set of observed variables is caused by a single latent parent (see Appendix for extensions). Next, we focus on identifying these latent variables.

Since each latent variable has certain observed children as its measured variables in the mixed LSM, one can identify these measured variables as causal clusters to determine the existence of latent variables. To ensure that latent variables are identified correctly and without redundancy, two issues need to be addressed:

(i). Finding all causal clusters from the observed variables and determining latent parents for each cluster.
(ii). Merging the causal clusters that share common latent parents to avoid redundancy.

For the first issue, we first define the concepts of causal clusters and one-factor (multi-factor) structures.

**Definition 4.9** (Causal cluster). In the Mixed LSM model, $C_i = \{X_1, \cdots, X_p\}$ is a causal cluster if and only if all variables in $\{X_1, \cdots, X_n\}$ share the common latent parent.

Due to the multi-factor model setting, we further define the observed set with a multi-factor structure.

**Definition 4.10** (One-factor & multi-factor cluster). For a causal cluster $C_i$, if $\forall X_i \in C_i$, $X_i$ has only one latent parent in the causal graph $\mathcal{G}$, $C_i$ is a one-factor cluster (or one-factor structure if $|C_i| = 1$). Otherwise, it is a multi-factor cluster (or multi-factor structure if $|C_i| = 1$).

To determine the number of latent parents in a causal cluster, it is crucial to first distinguish the multi-factor structure.

**Algorithm 1** Finding Causal Clusters

**Require:** Data set $\mathbf{X} = \{X_1, \ldots, X_m\}$
**Ensure:** The set of causal clusters $\mathbf{C}$ and its corresponding latent parent number set $\mathcal{L}$.

1: Initialize the causal cluster set $\mathbf{C} = \emptyset$;
2: Initialize the dictionary $\mathcal{L} = \emptyset$;
3: $\tilde{\mathbf{X}} \leftarrow$ Discretize all continuous variables in $\mathbf{X}$ by ranks' stop increasing criteria (Theorem 4.5);
4: $\mathbf{X}_c \leftarrow$ one-factor group by Proposition 4.11;
5: **for** $\forall X_i, X_j \in \mathbf{X}_c$, and $X_k \in \mathbf{X} \setminus \{X_i, X_j\}$ **do**
6:     **if** Proposition 4.13 hold for $\{X_i, X_j, X_k\}$ **then**
7:         $\mathbf{C} = \mathbf{C} \cup \{X_i, X_j, X_k\}$;
8:         $\mathcal{L} \leftarrow \{\{X_i, X_j, X_k\} : 1\}$ for recording the number of the latent parent of this causal cluster;
9:     **end if**
10: **end for**
11: $r \leftarrow$ the cardinality of latent support by Proposition 4.13;
12: **for** each set $C_i \subset \mathbf{X} \setminus \mathbf{X}_c$ **do**
13:     $\mathbf{C} \leftarrow$ multi-factor causal cluster by Proposition 4.15;
14:     $\mathcal{L} \leftarrow \{C_i : m\}$ with $m = \frac{\mathrm{Rk}(\mathbb{P}(\mathbf{X}_p))}{r}$;
15: **end for**
16: $\mathbf{C} \leftarrow$ merge the causal clusters that share the common latent parent by Proposition 4.16;
17: return $\mathbf{C}$ and $\mathcal{L}$;

---

Although the cardinality of the latent support is unknown, this structure can be identified by leveraging its graphical properties, i.e., tensor rank condition.

**Proposition 4.11** (Distinguish multi-factor structure). *For $X_i \in \mathbf{X}$, $X_i$ has only one latent parent if $\forall X_j \in \mathbf{X} \setminus \{X_i\}$, $\mathrm{Rank}(\mathbb{P}(X_i, X_j))$ is invariant. Otherwise, $X_i$ is caused by more than one latent parent in discrete LSM.*

Based on Proposition 4.11, one can divide the observed into two groups, one group is the observed variable that has only one latent parent, and another is the observed variable that has a multi-factor structure. Now, one can identify the cardinality of latent support by identifying the rank of the contingency table of any two observed in the first group. We first define such a group as the one-factor group.

**Definition 4.12** (One-factor group). Let $\mathbf{X}_c \subseteq \mathbf{X}$ denote a subset of observed variables, where each observed $X_i \in \mathbf{X}_c$ has exactly one latent parent (one-factor structure).

One can identify the causal clusters that share only one latent parent, as shown in the following result.

**Proposition 4.13** (One-factor cluster). *Let $\{X_i, X_j\} \subseteq \mathbf{X}_c$, and $X_k \in \mathbf{X} \setminus \{X_i, X_j\}$, $\{X_i, X_j, X_k\}$ is a causal cluster if $\forall X_s \in \mathbf{X} \setminus \{X_i, X_j, X_k\}$, $\mathrm{Rank}(\mathbb{P}(X_i, X_j, X_k, X_s)) = r$, where $r$ is the cardinality of latent support that can be identified by $\mathrm{Rank}(\mathbb{P}(X_i, X_j))$.*

*Example* 4.14. Take the Fig .2 as an example. One can see that $\{X_7, X_8, \tilde{X}_9\}$, $\{X_6, \tilde{X}_5, \tilde{X}_4\}$ and $\{\tilde{X}_{10}, \tilde{X}_{11}, \tilde{X}_{12}\}$ are identified as one-factor clusters.

Similarly, one can identify the causal cluster from $\mathbf{X} \setminus \mathbf{X}_c$ to determine the multi-factor structure.

**Proposition 4.15** (Multi-factor cluster). *Let $\{X_i, X_j, X_k\} \subseteq \mathbf{X} \setminus \mathbf{X}_c$, $\{X_i, X_j, X_k\}$ is a multi-factor causal cluster that caused by $n$ latent parents if $\forall X_s \in \mathbf{X} \setminus \{X_i, X_j, X_k\}$, (i) $\mathrm{Rank}(\mathbb{P}(X_p, X_q)) = r^n$, where $\forall p, q \in \{i, j, k\}$, and (ii) $\mathrm{Rank}(\mathbb{P}(X_i, X_j, X_k, X_s)) = r^n$.*

Once the causal cluster is identified, one can determine the existence of latent variables. However, it cannot ensure that the latent variable is identified without redundancy, e.g., two causal clusters that share the same latent parent. Thus, it is necessary to merge the causal clusters that share a common latent into one causal cluster (the second issue).

**Proposition 4.16** (Merge Rule). *Let $\mathbf{C}_1$ and $\mathbf{C}_2$ be two causal clusters identified by Proposition 4.13 or Proposition 4.15, then $\mathbf{C}_1$ and $\mathbf{C}_2$ share the common latent parent if one of the following conditions holds:*

*R1.* $\mathbf{C}_1 \cap \mathbf{C}_2 \neq \emptyset$, *or*
*R2.* *One of them is a multi-factor cluster, such as $\mathbf{C}_1$, and $\mathrm{Rank}(\mathbb{P}(\mathbf{C}_1 \cup \mathbf{C}_2)) = \mathrm{Rank}(\mathbb{P}(\mathbf{C}_1))$.*

Now, one can identify all latent variables by their causal clusters. The procedure is summarized in Algorithm 1.

### 4.2.2. STEP II: STRUCTURE LEARNING FOR LATENT VARIABLES

In Step I, we identify the causal clusters and determine the number of latent variables based on whether they belong to one-factor or multi-factor clusters. In this stage, we will show how the causal structure of latent variables can be identified up to a Markov equivalent class based on the identified clusters.

To this end, we follow the constraint-based method by (Chen et al., 2024) and present the conditional independence test among latent variables as follows.

**Theorem 4.17** (d-separation among latent varaible). *In the mixed LSM model, suppose the Markov condition, faithfulness assumption, full-rank condition, and completeness condition hold. Let $r$ be the dimension of latent support, given the discretized data $\mathbf{X}$, then $L_i \perp L_j | \mathbf{L}_p$ if and only if $\mathrm{Rank}(\mathcal{T}_{(X_i, X_j, \mathbf{x}_{p1}, \mathbf{x}_{p2})}) = r^{|\mathbf{L}_p|}$, where $X_i$ and $X_j$ are the pure children of $L_i$ and $L_j$, $\mathbf{X}_{p1}$ and $\mathbf{X}_{p2}$ are two disjoint child set of $\mathbf{L}_p$ that satisfy (i) $\forall X_i \in \mathbf{X}_{p1} \cup \mathbf{X}_{p2}$, $X_i$ is a sufficient measured variable and, (ii) $\forall L_i \in \mathbf{L}_p$, $\mathrm{Ch}_{L_i} \cap \mathbf{X}_{p1} \neq \emptyset$, $\mathrm{Ch}_{L_i} \cap \mathbf{X}_{p2} \neq \emptyset$.*

In Theorem 4.17, a sufficient measured variable $X_i$ refers to a variable that is either continuous or a discrete variable

with larger support than its latent parents.

*Example* 4.18. Consider the structure in Fig .2. Suppose that the cardinality of latent support is $r$. To test the CI relation $L_3 \perp\!\!\!\perp L_4 | \{L_1, L_2\}$, let $\mathbf{X}_p = \{\tilde{X}_9, \tilde{X}_{10}, \tilde{X}_1, \tilde{X}_2\}$, one can test the rank of the probability tensor $\mathbb{P}(\mathbf{X}_p)$, where $\tilde{X}_1$ and $\tilde{X}_2$ is discretized variables of $X_1$ and $X_2$. That is, $\text{Rank}(\mathbb{P}(\mathbf{X}_p)) = r^2 \Leftrightarrow L_3 \perp\!\!\!\perp L_4 | \{L_1, L_2\}$.

Based on Theorem 4.17, one can extend the PC-TENSOR-RANK algorithm (Chen et al., 2024) into the case of mixed observational data, by discretizing all continuous variables in $\mathbf{X}$ by ranks' stop increasing criteria (Theorem 4.5). Due to the space limitation, the complete algorithm is provided in Appendix. By this, we further give the identification result of mixed LSM.

**Theorem 4.19** (Identification of Mixed LSM). *In the mixed LSM, assuming the Markov condition, faithfulness assumption, full-rank condition, and completeness condition hold, if all latent variables have the same support and at least one set of observed variables is caused by a single latent parent, the causal structure of the latent variables can be identified up to a Markov equivalence class.*

## 5. Simulation Experiments

In this section, we conduct simulation experiments to verify the accuracy and effectiveness of our method. Baseline approaches include PC-TENSOR-RANK (abbreviated as TS-PC) (Chen et al., 2024), and mixed latent tree (abbreviated as Mixed-LT) (Zhou et al., 2020). To ensure fairness, since TS-PC can only handle discrete data, we directly discretize the continuous data using random cut points, and then apply TS-PC. For our algorithm, we test the matrix rank using the hypothesis test by (Mazaheri et al., 2023) and the tensor rank following (Chen et al., 2024).

**Setup.** In the simulation studies, we consider the different combinations of various structure models and measurement models. Specifically, for the structure model, we consider the following three typical cases: Case I: $L_1 \to L_2$; Case II: $L_1 \to L_2 \to L_3$; Case III: the structure of latent variables is shown in Fig. 2. For the measurement model, each latent variable has three pure observed variables, i.e., $L_i \to \{X_{3i+1}, X_{3i+2}, X_{3i+3}\}$. We randomly choose 1/3 of the observed variables to be continuous variables.

**Generation Process.** To ensure that the generated data satisfies the completeness condition, we use a mixture model as a practical and effective way to simulate data. Although the data is generated from a mixture model, the resulting joint distribution still reflects nonlinear dependencies (due to the probability property of the mixture model), and thus remains consistent with the setting described in Eq. (1). In all cases, the data generation process follows the following procedure: (i) we generate the probability contingency table

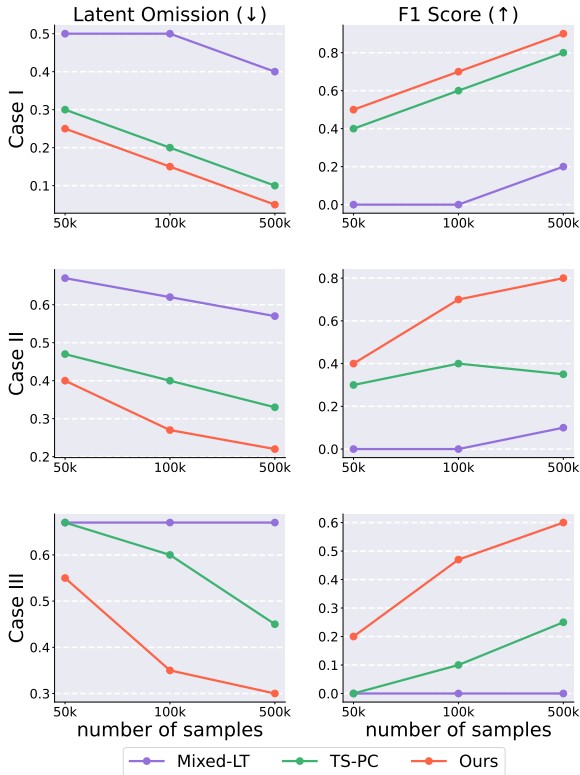

*Figure 3.* Performance under different setups.

of latent variables in advance, according to different latent structures (e.g., SM1), then (ii) we generate the conditional contingency table of discrete observed variables (condition on their latent parent) and each component of a Gaussian mixture model for continuous observed variables, and finally (iii) we sample the observed data according to the probability contingency table and Gaussian mixture model, where the dimension of latent support $r$ is set to 2 and the dimension of all discrete observed variables support is set to 3, sample size ranged from $\{5k, 10k, 50k\}$.

**Evaluation Metric.** For each simulation study, we randomly generate the dataset and apply the proposed algorithm and baselines to these data. We use latent omission to assess the performance of causal clusters from each algorithm, which can be referred to (Silva et al., 2006). Moreover, to assess the ability of these algorithms to discover the causal structure among latent variables, we use the F1 score as the evaluation metric. Each experiment was repeated ten times with randomly generated data.

**Results Analysis.** The results are reported in Fig. 3. Our method consistently delivers the best outcomes across most scenarios, demonstrating its capability to identify both the causal clusters and the causal structures of latent variables with mixed data types. In contrast, the TS-PC approach performs poorly, as random discretization may violate the full-rank assumption (see Appendix B). Additionally, the Mixed-

LT algorithm shows suboptimal performance in structure learning of latent variables due to their limitations to specific structural models, such as tree structures, or assumptions that latent variables are binary. More experimental results and discussions are provided in the Appendix.

## 6. Conclusion

In this paper, we extend the tensor rank condition to discrete latent structure models with mixed-type observational data. We demonstrate that, under the completeness condition, the tensor rank condition holds for discretized data, requiring only two sufficient measured observed variables. Based on this extension, we propose a structure learning algorithm that first identifies causal clusters to determine the number of latent variables and then infers the causal structure among them using their measured variables. This identification result extends the identification bounds of discrete latent structure models with a multi-factor structure. Future work includes allowing for impure observed structures and exploring more general non-linear models.

### Acknowledgements

Feng Xie would like to acknowledge the support by the National Natural Science Foundation of China (62306019). RC would like to acknowledge the support by by National Key R&D Program of China (2021ZD0111501), National Science Fund for Excellent Young Scholars (62122022), Natural Science Foundation of China (61876043, 61976052, 62476163), the major key project of PCL (PCL2021A12). ZF would like to acknowledge the support by the National Natural Science Foundation of China under grants No. 62476163 and U24A20233, and the Guangdong Basic and Applied Basic Research Foundation under grant number 2023B1515120020. KZ would like to acknowledge the support from NSF Award No. 2229881, AI Institute for Societal Decision Making (AI-SDM), the National Institutes of Health (NIH) under Contract R01HL159805, and grants from Quris AI, Florin Court Capital, and MBZUAI-WIS Joint Program. We would like to thank the anonymous reviewers for their helpful comments.

### Impact Statement

Learning causal structure among latent variables is essential throughout the data-driven sciences and has attracted much attention. Our research focuses on learning causal structure among discrete latent variables only from their measured variables, which can be continuous variables or discrete variables. This is typically encountered in fields like social sciences, economics, public health, and neuroscience. We assess the impact of our work in the context of these fields. However, the applicability of existing methods is often limited in practice, as the tensor rank condition relies on certain assumptions such as the full-rank condition or completeness condition. Notable merits of our work include establishing the connection between the tensor rank condition and the causal graph pattern, providing a practical method for constructing discretized data, and applying the tensor rank condition to learn the causal structure from purely observational data.

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

## A. Related Works

**Latent Confounders.** Unobserved confounding has long been a significant challenge for reliably drawing causal inferences and conducting statistical analyses from observational data. In the causal inference, most methods focus on addressing the problem of effect estimation in the presence of latent confounders, such as instrumental variables-based approaches (Brundy & Jorgenson, 1971; Viberg et al., 1997; Myers et al., 2011; Xie et al., 2022a; Rudolph et al., 2024), the control outcome-based methods (Tchetgen Tchetgen, 2014; Shi et al., 2020), confounding functions-based ones (Kasza et al., 2017; Miao et al., 2024). There are also some statistical models that account for latent confounders, such as those proposed in (Tofighi et al., 2019; Valente et al., 2017; Dziadkowiec, 2023; Liu & Wang, 2021). Most of these methods focus on the problem of estimation, rather than addressing the challenge of structure learning.

**Mixture Model.** Mixture models are closely related to our problem setup. Traditional methods focus on the parameter identifiability of mixture models, as seen in (Lindsay & Roeder, 1993; McLachlan & Peel, 2000; Allman et al., 2009; Anandkumar et al., 2014; Kim & Lindsay, 2015; Mena & Walker, 2015; Yang et al., 2020; Tahmasebi et al., 2018; Kargas & Sidiropoulos, 2019). To learn the causal structure in the presence of latent confounders, (Mazaheri et al., 2023; Anandkumar et al., 2012) present a rank-based approach to infer the $d$-separation relations among observed variables, leveraging the identifiability of mixture models. For learning the structure of latent variables, (Kivva et al., 2021) propose a mixture oracle-based method to recover the distribution of latent variables and then perform causal discovery among them. However, the identification results depend on the existence of a mixture oracle, and it remains unclear how such an oracle can be identified.

**Latent Variable Model.** Latent variable graphical models have been well studied in the literature. Most work focuses on linear models, such as the linear latent variable model with Gaussian noise assumption (Silva et al., 2006; Kummerfeld & Ramsey, 2016; Kummerfeld et al., 2014; Huang et al., 2022), non-Gaussian linear latent structure models (Cai et al., 2019; Xie et al., 2020; 2022b; Chen et al., 2023; 2022; Jin et al., 2023), and copula model-based approaches (Cui et al., 2018). Less is known regarding structure learning between latent variables for nonlinear models. One typical method is based on non-linear ICA, such as (Zheng et al., 2022; Hyvarinen & Morioka, 2017; Hyvarinen et al., 2019). In the discrete model domain, there are a few methods, including the latent tree model (Choi et al., 2011; Mourad et al., 2013) and pure measurement model (Gu, 2022; Gu & Dunson, 2023; Chen et al., 2024). However, these methods focus on restricted latent structures or linear assumptions and cannot handle discrete latent variable structures with mixed-type observational data.

## B. More Details on the Motivating Example for Discretization

In this section, we discuss two key questions: (i) why it is necessary to design a method for discretization, and (ii) the detailed implementation of the discretization process. The first part highlights that without a careful discretization strategy, the tensor rank condition may fail to hold in the discretized data. The second part provides an illustrative example to show this procedure in practice.

We begin by discussing the first question. When we learn the causal structure using the tensor rank condition, a key assumption for applying the tensor rank condition is the full-rank assumption (Chen et al., 2024), which stipulates that the conditional probability table must have full rank. However, when continuous data are discretized arbitrarily, this assumption may be violated, potentially undermining the validity of the tensor rank condition. For example, in Fig .4, one can see that there exists a discretization of $X_i$ such that $\mathbb{P}(\tilde{X}_i|L)$ is not full rank.

In this example, we simulate a simple structure $L \to X_i$ where $X_i$ is a continuous variable and $L$ is a binary variable. Our goal is to discretize $X_i$ into a discrete variable $\tilde{X}_i$ with support $\{0, 1, 2, 3\}$. Thus, To achieve this, we randomly select three cut points $\{\frac{\pi}{2}, \pi, \frac{3\pi}{2}\}$, on the marginal distribution of $X_i$ (i.e., the pink area). When examining the conditional distributions $\mathbb{P}(X_i|L = 0)$ and $\mathbb{P}(X_i|L = 1)$, represented by the blue and green areas, respectively, we observe that the conditional probability table is not full rank. This is because the blue and green areas are identical in each partition, thereby violating the full-rank condition. Therefore, if we discretize the continuous variable arbitrarily, the structure learned by the tensor rank condition may be incorrect. This is the reason why we need to present the ranks stopped increasing criteria (Theorem 4.5) during the discretization process.

*Remark* B.1. In (Kargas & Sidiropoulos, 2019), it is demonstrated that the identifiability of mixture distributions holds even after discretization, and a practical algorithm is provided to learn the conditional probability by tensor decomposition (actually, the tensor rank) for discretized data. Based on their algorithm, one can efficiently estimate the tensor rank of the

discretized probability tensor.

Next, we provide additional implementation details for the discretization methodology. Specifically, we describe how to discretize the data such that the full-rank condition holds. We generate data using the following directed acyclic graph (DAG): $L \to X$, $L \to Y$, $L \to Z$, where $X$ and $Y$ are continuous variables, and $Z$ is a binary discrete variable. The latent variable $L$ has the support $\{0, 1\}$. The marginal distribution of $X$ and $Y$ is a mixture of the Laplace distribution (Figure 5 (a) and (c)). Through proper discretization of $X$ and $Y$ into $\tilde{X}$ and $\tilde{Y}$ (Figure 5 (b) and (d)), respectively, for example, selecting cutpoints $2.06$ and $0.26$ for $X$, and cutpoints $-0.94$ and $0.83$ for $Y$, we discretize the two continuous variables into two discrete variables $\tilde{X}$ and $\tilde{Y}$, as shown in Figure 5 (b) and (d). In this procedure, we maximize $\mathrm{Rank}(\mathbb{P}(\tilde{X}, \tilde{Y}))$, ensuring that $\mathbb{P}(\tilde{X}|L)$ and, hence, $\mathbb{P}(\tilde{Y}|L)$ satisfy the full-rank assumption. For instance, the joint distribution $\mathbb{P}(X, Y, Z = i)$ with $i = 0, 1$ is shown in Figure 6. We discretize this distribution into a discrete joint probability table, as shown in Figure 7, which is a rank-2 tensor, indicating that the conditional probability table is full rank. Based on these results, one can use $\tilde{X}$, $\tilde{Y}$, and $Z$ to compute the tensor rank of the probabilistic contingency table and recover the support of $L$.

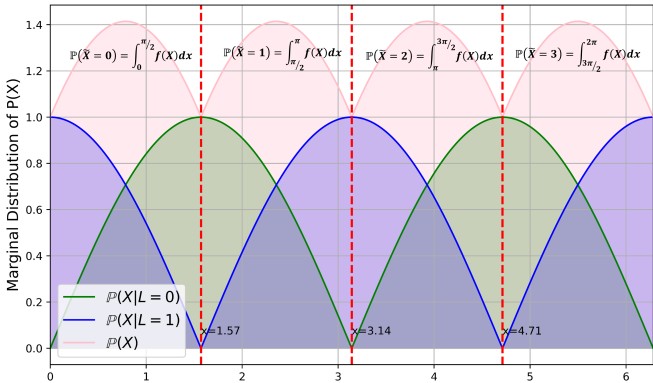

*Figure 4.* Example for discretized data, where the latent parent $L$ of $X_i$ has $\mathrm{supp}(\mathrm{L}) = \{0, 1\}$. There are three cut point $\{\frac{\pi}{2}, \pi, \frac{3\pi}{2}\}$ that lead to $\mathbb{P}(\tilde{X}_i)$ with support $d_i = \{0, 1, 2, 3\}$. One can see that $\mathbb{P}(\tilde{X}_i|L)$ is not full rank because the green and blue areas are equal in each partition.

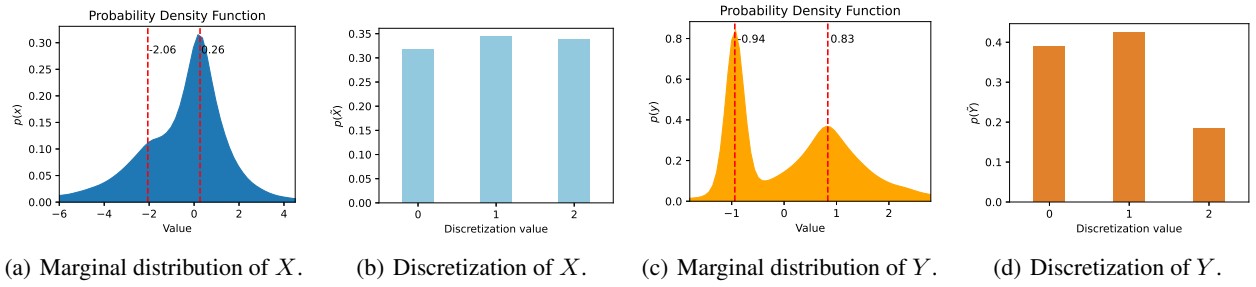

| (a) Marginal distribution of $X$. | (b) Discretization of $X$. | (c) Marginal distribution of $Y$. | (d) Discretization of $Y$. |

*Figure 5.* Exanple for illustrating the procedure of discretization. For marginal distribution $X$ and $Y$ (i.e., subfig (a) and subfig (c)), we discretize them into two discretized variables (i.e., subfig (b) and subfig (d)).

## C. More Details on PC-TENSOR RANK Algorithm

The specific details of the PC-TENSOR RANK algorithm are provided below.

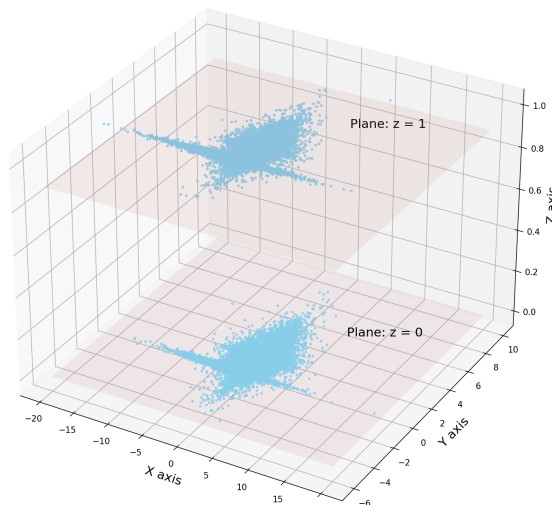

3D Scatter Plot

| $p(\tilde{X}, \tilde{Y}, Z = 0)$ | $\tilde{Y} = 0$ | $\tilde{Y} = 1$ | $\tilde{Y} = 2$ |
|---|---|---|---|
| $\tilde{X} = 0$ | 0.170 | 0.169 | 0.001 |
| $\tilde{X} = 1$ | 0.116 | 0.133 | 0.081 |
| $\tilde{X} = 2$ | 0.116 | 0.133 | 0.081 |

| $p(\tilde{X}, \tilde{Y}, Z = 1)$ | $\tilde{Y} = 0$ | $\tilde{Y} = 1$ | $\tilde{Y} = 2$ |
|---|---|---|---|
| $\tilde{X} = 0$ | 0.145 | 0.142 | 0.001 |
| $\tilde{X} = 1$ | 0.114 | 0.137 | 0.105 |
| $\tilde{X} = 2$ | 0.114 | 0.137 | 0.105 |

*Figure 7.* Probabilistic contingency table of $\mathbb{P}(X, Y, Z = i)$ with $i = 0, 1$. One can see that $\mathrm{Rank}(\mathbb{P}(\tilde{X}, \tilde{Y}, Z = 0)) = 2$ and $\mathrm{Rank}(\mathbb{P}(\tilde{X}, \tilde{Y}, Z = 1)) = 2$, indicating $L$ (with $|\mathrm{supp}(\mathbb{P}(L))| = 2$) $d$-separates $X, Y$ and $Z$.

*Figure 6.* Joint distribution of $\mathbb{P}(X, Y, Z = i)$ with $i = 0, 1$.

---

**Algorithm 2** PC-TENSOR-RANK

---

**Input**: The discretized data set $\mathbf{X} = \{X_1, \ldots, X_m\}$ and causal cluster $\mathcal{C}$
**Output**: A partial DAG $\mathcal{G}$.

1: Initialize the maximal conditions set dimension $k$;
2: Let $L_i$ denote as $C_i, C_i \in \mathcal{C}$;
3: Form the complete undirected graph $\mathcal{G}$ on the latent variable set $\mathbf{L}$;
4: **for** $\forall L_i, L_j \in \mathbf{L}$ and adjacent in $\mathcal{G}$ **do**
5:     *//Test the CI relations among latent variables by Theorem 4.17*
6:     **if** $\exists \mathbf{L}_p \subseteq \mathbf{L} \setminus \{L_i, L_j\}$ and $(|\mathbf{L}_p| < k)$ such that $L_i \perp\!\!\!\perp L_j | \mathbf{L}_p$ hold **then**
7:         delete edge $L_i - L_j$ from $G$;
8:     **end if**
9: **end for**
10: Search V structures and apply meek rules (Meek, 1995).
11: **return** a partial DAG $\mathcal{G}$ of latent variables.

---

## D. Analysis the Identification of Discrete Latent Variable Models under Sparsity Condition

We now extend the results of the previous section when there are some direct edges between the observed variable (impure structure) in contrast to the three-pure children condition. Our goal is to explore under what milder conditions the discrete latent variable model remains identifiable. Under more general sparsity assumption, we provide the identification results of the discrete latent variable model with the *discrete observed variables*. For continuous observed variables, the identification result can be easily extended by appropriately estimating the conditional distribution.

In the following, we aim to relax the purity assumption and the three-pure children assumption in the discrete latent variable model. The alternative identification condition is the sparsity assumption.

**Assumption D.1** (Sparsity assumption). each latent variable set $\mathbf{L}_p \subset \mathbf{L}$, in which every latent variable directly causes the same set of observed variables, has at least three observed children variable $X_i, X_j, X_k \in \mathbf{X}$ such that (i) $\{X_i, X_j, X_k\} \perp\!\!\!\perp \mathbf{X} \setminus \{X_i, X_j, X_k\} | \mathbf{L}_p$, and (ii) there are $\mathbf{X}_q \subset \mathbf{X} \setminus \{X_i, X_j, X_k\}$, $X_i \perp\!\!\!\perp X_j \perp\!\!\!\perp X_k | \{\mathbf{L}_p, \mathbf{X}_q\}$.

*Example* D.2. For example, for the $L_3$ in Fig .8, one have $X_{13}, X_{15}$ and $X_7$ satisfy the spasity assumption. Since given $X_{14}$, we have $X_{13} \perp\!\!\!\perp X_{15} \perp\!\!\!\perp X_7 | \{X_{14}, L_3\}$. By the tensor rank condition, the joint distribution $\mathbb{P}(X_{13}, X_{15}, X_7 | X_{14})$ follows the graphical implication.

Under the sparsity assumption, we allow edges between observed variables and do not impose the three pure children structure constraints. This leads to a more general identification result for discrete latent variable models. We define such a general discrete latent variable model as the *Sparse Discrete Latent Variable Model*.

**Definition D.3** (Sparse Discrete Latent Variable Model). A nonlinear causal model with latent confounders and its corresponding causal graph $\mathcal{G}$ is a Sparse Discrete Latent Variable Model (Discrete Sparse-LSM) if all observed variables are discrete and satisfy the following conditions:

- (Sparsity Assumption) each latent variable set $\mathbf{L}_p \subset \mathbf{L}$, in which every latent variable directly causes the same set of observed variables, satisfy the sparsity assumption (Assumption D.1);

- (Two-Sufficient Measurement Assumption) each latent variable set $\mathbf{L}_p \subset \mathbf{L}$, in which every latent variable directly causes the same set of observed variables, has at least two sufficient measured variables, i.e., larger support than their latent parents.

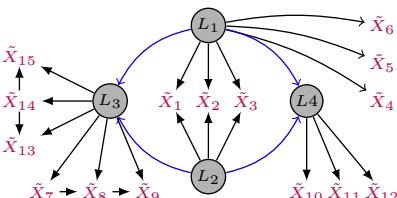

*Figure 8.* A generalized discrete latent structure in the presence of mixed-type observed variables, where $X_i$ represents continuous observed variables $\tilde{X}_i$ represents discrete observed variables (purple), and $L_i$ represents a discrete latent variable.

To ensure the cardinality of latent support is identifiable, we require the following assumption.

**Assumption D.4.** All latent variable has the same support, and there exists at least one observed variable set $\mathbf{X}_p$ with $|\mathbf{X}_p| \geq 3$, $\mathbf{X}_p$ has only one latent parent.

*Remark* D.5 (Discussion on assumption D.4). Assumption D.4 is only a sufficient condition for identifying the cardinality of the latent support. In (Mazaheri et al., 2023), the cardinality of the latent support is assumed to be known, while in (Kivva et al., 2021), the Subset condition is assumed. The Subset condition requires that the neighborhoods of two distinct latent variables do not overlap, thereby ensuring the identification of the components in the mixture model. Actually, it is possible to relax the Assumption D.4 to a more general case where the support of latent variable can be different, by using the minimal state space criteria (Chen et al., 2024).

In the following, we develop an identification algorithm that identifies the latent causal structure under the Discrete Spare-LSM. We still follow the strategy that first determines the latent variable by causal cluster, and then infer the causal structure based on these clusters. Since there are edges between observed variables (called *impure structure*), it is necessary to find these impure structures to ensure the causal cluster can be identified correctly. In the discrete sparse-LSM, the impure structure can be identified by performing the conditional independent test in the presence of a latent confounder.

#### D.1. Conditional Independent Test in the presence of Latent Confounders

We begin with an illustrative example that demonstrates the connection between the tensor rank condition and the $d$-separation relations among observed variables in an impure structure. Take the structures in Fig .9 as an example, one can see that, $X_1 \perp\!\!\!\perp X_3 | \{X_2, L_1\}$ hold for two structures. Suppose $X_1, X_2, X_3$ has the same support $\{0, 1, 2\}$ and the latent variable $L_1, L_2$ has the same support $\{0, 1\}$, let $\mathbb{P}(X_1, X_3 | X_2 = c)$ be the conditional probability contingency table, denote $\tilde{p}_{j|c,i} = \mathbb{P}(X_1 = j | X_2 = c, L_1 = i)$, $\hat{p}_{j|c,i} = \mathbb{P}(X_3 = j | X_2 = c, L_1 = i)$, and $p_{i|c} = \mathbb{P}(L_1 = i | X_2 = c)$, under the Markov assumption, faithfulness assumption, one can see that

$$\mathbb{P}(X_1, X_3 | X_2 = c) = \underbrace{\begin{bmatrix} \tilde{p}_{0|c,0} & \tilde{p}_{0|c,1} \\ \tilde{p}_{1|c,0} & \tilde{p}_{1|c,1} \\ \tilde{p}_{2|c,0} & \tilde{p}_{2|c,1} \end{bmatrix}}_{\mathbb{P}(X_1 | X_2 = c, L_1)} \cdot \underbrace{\begin{bmatrix} p_{0|c} & \\ & p_{1|c} \end{bmatrix}}_{\text{Diag}(\mathbb{P}(L_1 | X_2 = c))} \cdot \underbrace{\begin{bmatrix} \hat{p}_{0|c,0} & \hat{p}_{1|c,0} & \hat{p}_{2|c,0} \\ \hat{p}_{0|c,1} & \hat{p}_{1|c,1} & \hat{p}_{2|c,1} \end{bmatrix}}_{\mathbb{P}^{\intercal}(X_3 | X_2 = c, L_1)}. \tag{3}$$

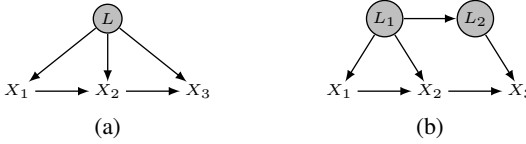

*Figure 9.* Example of the impure structure that can be identified by tensor rank condition.

Under the non-degenerate assumption (i.e., the conditional contingency table is full rank and non-zero elements in the diagonal matrix), one can see that $\mathrm{Rank}(\mathbb{P}(X_1, X_3|X_2 = c)) = 2$. This implies that when considering the conditional distribution of $X_1$ and $X_3$ given the observed variable set $X_2$ (which $d$-separates $X_1$ from $X_3$ by combining with the latent variable $L_1$), the rank of the conditional tensor is determined by the latent parent $L_1$. Thus, one can test the conditional independence relations among observed variables by identifying these properties, which can help detect the impure structure among the measured variables.

**Theorem D.6** (CI relations among observed variables)**.** *In the Discrete Sparse-LSM, suppose the Markov assumption, faithfulness condition, and the non-degenerate condition hold. Let $r$ denote the cardinality of latent support, $X_i \perp\!\!\!\perp X_j|\{\mathbf{X}_p, L\}$ hold if and only if for the conditional probability tensor $\mathrm{Rank}(\mathbb{P}(X_i, X_j|\mathbf{X}_p)) = r$.*

*Proof.* Observe that any two observed variables (if they are purely measured variables) are separated by one of their latent parents in the discrete sparse-LSM model. This result can be directly proven by combining the non-degenerate condition with the findings from (Anandkumar et al., 2012). □

By theorem D.6, one can find the general pure child set for each latent variable by designing a proper search algorithm.

**Lemma D.7** (Stop decreasing for probability matrix)**.** *For any pair of observed variables $X_i$ and $X_j$, let $L_i$ and $L_j$ be their latent parent. If there is no $\mathbf{X}_p$ such that $\mathrm{Rank}(\mathbb{P}(X_i, X_j|\mathbf{X}_p))$ has a lower rank than any $\mathrm{Rank}(\mathbb{P}(X_i, X_j|\mathbf{X}_q)) = r$, then $\min(|\mathrm{supp}(\mathbb{P}(L_i))|, |\mathrm{supp}(\mathbb{P}(L_j))|) = r$.*

*Proof.* The proof is straightforward. Suppose $|\mathrm{supp}(\mathbb{P}(L_i))| < |\mathrm{supp}(\mathbb{P}(L_j))|$, based on Theorem D.6, if we find the conditional set $\mathbf{X}_q$ in which $\{L_i, \mathbf{X}_q\}$ is the conditional set that $d$-separates any pair variable in $\mathbf{X}_p$, then $\mathrm{Rank}(\mathbb{P}(X_i, X_j|\mathbf{X}_q)) = r$, under the Markov assumption, faithfulness assumption, and the non-degenerate condition. Otherwise, $\mathrm{Rank}(\mathbb{P}(X_i, X_j|\mathbf{X}_q)) > r$ according to the graphical implication of tensor rank condition. Therefore, if we find that $\mathrm{Rank}(\mathbb{P}(X_i, X_j|\mathbf{X}_q))$ is minimal (do not exist $\tilde{\mathbf{X}}_q$ such that the conditional probability tensor has lower rank), then $\{L_i, \mathbf{X}_q\}$ is the conditional set with the minimal cardinality of latent support, i.e., $\min(|\mathrm{supp}(\mathbb{P}(L_i))|, |\mathrm{supp}(\mathbb{P}(L_j))|) = r$. □

Based on Lemma D.7, all $d$-separations among observed variables can be identified using an appropriate search algorithm, such as (Anandkumar et al., 2012; Mazaheri et al., 2023). Moreover, existing methods can identify whether an observed variable set is caused by a latent confounder, such as (Silva et al., 2006; Chen et al., 2021a). Thus, one can determine the observed variable set in which each observed variable has at least one latent parent. We omit the specific algorithm and assume that all $d$-separation relations between any pair of measured variables $X_i, X_j$ are recorded in a set denoted as dset. For instance, $\mathrm{dset}(X_i, X_j) = \mathbf{X}_p$ implies that there exist $\exists \mathbf{L}_q \subseteq \mathbf{L}$, such that $\mathbf{L}_q \cup \mathbf{X}_p$ $d$-separates $X_i$ from $X_j$.

### D.2. Identifying the Causal Structure of Discrete Sparse-LSM

Based on the fact that all $d$-separation relations among observed variables are identifiable, we can infer the identifiability of the discrete sparse latent variable model. Since its identification algorithm is hybrid and lacks elegance (e.g., a hybrid algorithm from the modified PC-algorithm (Mazaheri et al., 2023), the FINDPATTERN algorithm (Silva et al., 2006) and the discrete latent structure learning algorithm (Chen et al., 2024)), we primarily focus on discussing its identifiability result.

**Theorem D.8** (Identification of discrete sparse-LSMs)**.** *Given an unbiased estimation of the conditional distribution of any observed variables, in the discrete sparse latent structure model, suppose the Markov assumption, faithfulness condition, non-degenerate condition, and assumption D.4 hold. The causal structure of the latent variable can be identified up to a Markov equivalence class.*

*Proof.* To complete this proof, there are two stages of identification that need to be discussed: the causal cluster that determines the latent variable and the structure model among latent variables.

Before providing the proof, we first demonstrate that all observed variable sets with at least one latent parent can be identified using the following corollary.

**Corollary D.9.** *For $X_i$ and $X_j$, if one of them does not have a latent parent, then there exists a subset $\mathbf{X}_q \subseteq \mathbf{X} \setminus \{X_i, X_j\}$, such that $\mathrm{Rank}(\mathbb{P}(X_i, X_j | \mathbf{X}_q)) = 1$.*

Since one of $X_i$ and $X_j$ does not have a latent parent, then $X_i$ and $X_j$ are $d$-separated by the observed variable set. According to Remark G.1 in (Chen et al., 2024), this result is proven. Based on this corollary, all observed variables that do not have a latent parent can be removed. Now, we can continue the proof of discrete sparse-LSM.

Stage I: Identify the cardinality of latent support. Based on Lemma D.7 and Assumption D.4, one can identify the cardinality of latent support, i.e., identifying the minimal rank for the (conditional) probability table over two observed variables that are caused by a single latent parent. We can iterate over all pairs of variables and identify the cardinality of latent support by finding the minimum rank constraint of the probability contingency table.

Stage II: Determine the latent variable. Under the sparsity assumption and the result of Theorem D.6, one can see that, $\{X_i, X_j, X_k\}$ is a causal cluster, if $\mathrm{Rank}(\mathbb{P}(\mathbf{X}_p | \mathbf{X}_q)) = r$ with $\mathbf{X}_p = \{X_i, X_j, X_k, X_s\}$ for any $X_s \in \mathbf{X} \setminus \{X_i, X_j, X_k\}$, and $\mathbf{X}_q \cup L_i$ ($L_i$ is the latent parent of them) is the conditional set that $d$-separates any pair variable in $\{X_i, X_j, X_k\}$, where $\mathbf{X}_p \cap \mathbf{X}_q = \emptyset$. The reason is as follows.

First, by Theorem D.6, one can get all conditional independent relations between any pair of observed variables (Mazaheri et al., 2023; Anandkumar et al., 2012). We can record these $d$-separation sets for each pair of observed variables $X_i$ and $X_j$, denoted by $\mathrm{dset}(X_i, X_j) = \mathbf{X}_p$ where $X_i \perp\!\!\!\perp X_j | \{L_i, \mathbf{X}_p\}$, $L_i$ is one of the parent variables of $X_i$ or $X_j$. For an observed variable set $\{X_i, X_j, X_k\}$ and any $X_s \in \mathbf{X} \setminus \{X_i, X_j, X_k\}$, let $\mathbf{X}_q = \bigcup_{i,j \in \{i,j,k,s\}} \mathrm{dset}(X_i, X_j)$, i.e., the combination of the $d$-separation set for any pair variable in $\mathbf{X}_p$. Therefore, if $\{X_i, X_j, X_k\}$ share the one latent parent variable set, denoted by $L_t$, then one can see that $\{L_t\} \cup \mathbf{X}_q$ is a conditional set that $d$-separates any pair variable in $\mathbf{X}_p$. Similar to the proof of (Chen et al., 2024), one can see that $\{L_t\} \cup \mathbf{X}_q$ is the minimal conditional set with support $r$. Thus, $\mathrm{Rank}(\mathbb{P}(\mathbf{X}_p | \mathbf{X}_q)) = r$.

Second, by repeating the above procedure, the causal cluster can be identified. Moreover, since the cardinality of the latent support is identifiable, let $r$ represent the cardinality of the latent support. Without loss of generality, the number of latent parents can be determined as $\frac{nr}{r}$, where $nr$ is the rank of probability tensor $\mathbb{P}(X_i, X_j, X_k, X_s | \mathbf{X}_q)$.

Third, we aim to show that the causal cluster can be further merged if they share a common latent parent, to avoid the redundant introduction of latent variables. Under the sparsity assumption and the result of $d$-separation for any pair observed variable, if for two causal clusters $\mathbf{C}_1$ and $\mathbf{C}_2$, $\mathbf{C}_1 \cap \mathbf{C}_2 \neq \emptyset$, then $\mathbf{C}_1$ and $\mathbf{C}_2$ share the common latent parent, according to the result of (Chen et al., 2024) (Proposition 4.5). Moreover, one can modify the result of Proposition 4.16 to check more merging rules. Let $\mathbf{X}_q$ be the conditional set for any pair variable in $\mathbf{C}_1$ and $\mathbf{C}_2$, by constructing the conditional probability tensor, Proposition 4.16 holds. By detecting the causal clusters that share a common latent parent and merging them into one causal cluster, the number of latent variables is identifiable.

Stage III: Identify the causal structure among latent variables, given the causal cluster for each latent variable. In stage II, we have recorded the $d$-separation set for any pair of observed variables. Thus, one can construct the conditional probability tensor $\mathbb{P}(\mathbf{X}_p | \mathbf{X}_q)$ for any set $\mathbf{X}_p \subseteq \mathbf{X}$ such that any pair variable $X_i, X_j \in \mathbf{X}_p$ are $d$-separated by $\{\mathbf{X}_q \cup L_i\}, L_i \in \mathbf{L}$. This means that, one can find that $\mathbf{L}_t \subseteq \mathbf{L}$ such that $\mathbf{L}_t \cup \mathbf{X}_q$ $d$-separates any pair variable in $\mathbf{X}_p$ and there do not exist $\tilde{\mathbf{L}}_t$ with $|\mathrm{supp}(\mathbb{P}(\mathbf{L}_t))| < |\mathrm{supp}(\mathbb{P}(\tilde{\mathbf{L}}_t))|$ also be the conditional set. According to the result of (Chen et al., 2024), the conditional independent relations among the latent variables are identified. Using the PC algorithm with latent variable (Chen et al., 2024), the causal structure among latent variables is identified up to a Markov equivalent class.

$\square$

### D.3. Discussion of Two Pure-children Condition

We aim to demonstrate that the measurement model is identifiable under the condition that each latent variable has only two pure measured variables, assuming the latent structure is fully connected. This result, which is also shown by (Gu, 2022), is not surprising. However, compared to (Gu, 2022), our identification algorithm is simpler and more efficient.

We only need to check the rank condition of a three-way tensor, to identify the causal cluster.

**Theorem D.10.** *In the discrete sparse-LSM, suppose the Markov assumption, faithfulness assumption, non-degenerate condition, and the assumption D.4 hold. When each latent variable has at least two pure measured variables and the latent structure is fully connected, the measurement model is identifiable.*

*Proof.* To identify the causal cluster by tensor rank condition, one can check for any $X_i, X_j \in \mathbf{X}, \forall X_k \in \mathbf{X} \setminus \{X_i, X_j\}$, let $\mathbf{X}_p = \{X_i, X_j, X_k\}$, if $\mathrm{Rank}(\mathbb{P}(\mathbf{X}_p)) = r$, then $\{X_i, X_j\}$ is a causal cluster, where $r$ is the cardinality of latent support. The proof is derived from the Proof of Proposition 4.3 in (Chen et al., 2024). That is, if $X_i, X_j$ are not a causal cluster, e.g., $L_2$ is the latent parent of $X_i$ and $L_1$ is the latent parent of $X_j$, then there exists $X_k \in \mathbf{X} \setminus X_i, X_j$ such that the $\mathrm{Rank}(\mathbb{P}(\mathbf{X}_p)) \neq r$ (see Fig 4 in (Chen et al., 2024)). Thus, all causal clusters can be identified by checking the rank constraints over the three-way probability tensor. Moreover, by applying the merging rule in Proposition 4.16, two causal clusters that share a common latent parent can be merged to avoid the redundant introduction of latent variables. Therefore, the identification of the measurement model is completed. $\qquad\square$

*Remark* D.11 (Discussion on two-sufficient measurement). It is important to contrast this condition with the sufficient observation assumption in (Chen et al., 2024), where the cardinality of each discrete observed variable is larger than the cardinality of the latent variable. One may note that in the mixed LSM, each latent variable has at least three observed children, but only requires two of them to be continuous variables, regardless of whether the third variable is discrete or continuous. Additionally, if the third variable is discrete, we do not assume any specific cardinality of its support; for instance, it can be a binary variable.

## E. More Details on the Difference between Mixture Model and Tensor Rank Condition

To learn discrete latent structures solely from observed variables, a traditional approach treats structural identification as a parameter identification problem in mixture models. Numerous studies focus on this approach, including (Gu, 2022; Kivva et al., 2021; Anandkumar et al., 2012). We are particularly interested in discussing whether the identification results based on mixture models offer more general identifiability than those achieved through tensor rank conditions.

In (Gu, 2022), it is shown that for a binary latent structure model, one can vectorize $\mathbb{P}(L_1, \cdots, L_m)$ to a marginal distribution $\mathbb{P}(L_v)$ that the support of $L_v$ is $\prod_{i=1}^{m} \mathrm{supp} |\mathbb{P}(L_i)|$. According to Kruskal's Theorem (Kruskal, 1977), the parameters of $P(X_i|L_v)$ and $\mathbb{P}(L_v)$ can be uniquely recovered through tensor decomposition, under the assumption that each latent variable has at least three pure children. Based on the identified parameter matrix, the measurement model becomes identifiable. In (Kivva et al., 2021), it is shown that the measurement model can be identified under a weaker structural condition, leveraging the identification results of a mixture model (Mixture Oracle). This is because the structure can be identified under weaker conditions by using a simplified version of Kruskal's condition, as proposed by Lovitz and Petrov (Lovitz & Petrov, 2023), as shown below.

**Lemma E.1** (Generalization of Kruskal's theorem (Lovitz & Petrov, 2023)). *Let $n \geq 2$ and $m \geq 3$ be integers, let $\mathcal{V} = \mathcal{V}_1 \otimes \cdots \mathcal{V}_m$ be a vector space over a field $\mathbb{F}$ and let*

$$x_{a,1} \otimes \cdots \otimes x_{a,m} : a \in [n] \subseteq \mathcal{V} \setminus \{0\} \tag{4}$$

*be a multiset of product tensors. For each $a \in [n]$, let $x_a = x_{a,1} \otimes \cdots \otimes x_{a,m}$. For each subset $S \subseteq [n]$ and index $j \in [m]$, let*

$$d_j^S = \dim \mathrm{span}\{x_{a,j} : a \in S\}. \tag{5}$$

*If $2|S| \leq \sum_{j=1}^{m}(d_j^S - 1) + 1$ for every subset $S \subseteq [n]$ with $2 \leq |S| \leq n$, then $\sum_{a \in [n]} x_a$ constites a unique tensor rank decomposition.*

Lemma E.1 shows that the conditions for unique tensor decomposition can be relaxed to a more general case. Specifically, by constructing a tensor that encapsulates structural information, Lemma E.1 enables a more broadly applicable identification result for the measurement model.

In summary, both (Gu, 2022) and (Kivva et al., 2021) use the unique tensor decomposition to identify the measurement model, relying on Kruskal's theorem or its generalized form.

# F. More Details on Completeness Condition

Here, we present some existing results that illustrate the generality of the completeness condition, primarily based on the findings from (Hu & Shiu, 2018).

*Remark* F.1 (Sufficient Condition for $f_{X|L}(x|l)$). The sequence $\{f_{X|L}(x|l)\}$ corresponding to a sequence $\{l : 1, 2, \cdots, r\}$ is linearly independent if one of the following conditions hold:

1) $\sum_{l=1}^{r} c_i f(x|l) = 0$ for all $x$ implies $c_i = 0$ for all $i = 1, 2, \cdots, r$;

2) for all $l$, $\lim_{x \to \infty} \frac{f(x|l+1)}{f(x|l)} = 0$ or $\lim_{x \to x_0} \frac{f(x|l+1)}{f(x|l)} = 0$ for some $x_0$.

In other words, many conditional densities satisfy the completeness condition. An important family of conditional distributions that exhibit completeness is the exponential family, including Gaussian, Poisson, Binomial, and certain multivariate forms of these. Another family that implies completeness consists of convolution density functions (Hu & Shiu, 2018). Additionally, (Newey & Powell, 2003) demonstrates that the semiparametric exponential family also satisfies the completeness condition. A detailed discussion of the completeness condition can be found in the supplementary material [4] of (Ying et al., 2023).

# G. Proof and Illustrations

**Notations.** Suppose $X, L, \tilde{X}$ are random variables with domains $\mathcal{X}, \mathcal{L}, \tilde{\mathcal{X}}$. We denote the probability law of $X$ as $\mathcal{L}_X := \mathbb{P} \circ X^{-1}$ and denote the cumulative density function (CDF) as $F_X$. For discrete variables $\tilde{X}$, we denote their levels as $l_{\tilde{X}}, l_L$, etc. We use the the column vector $\mathbb{P}(\tilde{X}|x) := [p(\tilde{x}_1|x), ..., p(\tilde{x}_{l_X}|x)]^\top$, the row vector $\mathbb{P}(x|\tilde{X}) := [p(x|\tilde{x}_1), ..., p(x|\tilde{x}_{l_x})]$, and the matrix $\mathbb{P}(X|L) := [\mathbb{P}(X|l_1), ..., \mathbb{P}(X|l_{l_L})]$ to represent their transition probabilities. For a continuous variable $X$, we denote the probability density function (PDF) as $f_X$ and the conditional PDF of $x$ given $L = l_r$ as $f_{x|l_r}$ ( or $f(X|L=l_r)$). We denote $\{\mathcal{X}_i\}_{l_X}$ a disjoint partition of $\mathcal{X}$ with $l_X$ subspaces, such that $\cup_i^{l_X} \mathcal{X}_i = \mathcal{X}$ and $\mathcal{X}_i \cap \mathcal{X}_j = \emptyset$ for $i \neq j$. We thereby define the discretized version of $X$ as $\tilde{X}$, and denoting $\tilde{X} := \tilde{x}_i$ to represent the event $X \in \mathcal{X}_i$ for any $i$. We denote $F(x|l_r)$ as the conditional CDF of $x$ given $L = l_r$. Besides, we denote $F(x|\boldsymbol{l}_{[r]}) := [F(x|l_1), ..., F(x|l_r)]^\top$, with $[r] := \{1, ..., r\}$. We let $A_1 \backslash A_2 := A_1 \cap A_2^c$ for two sets $A_1, A_2$. We denote $n$ as the sample size.

To simplify notation in the proof and distinguish $X_i$ and $X_j$ without using subscripts, we sometimes replace $X_i, X_j$ with $X, Y$. This implies that $Y$ is also an observed variable measured in relation to the latent variables in the mixed-LSM.

## G.1. Proof of Theorem 3.4

We first give a lemma that gives a specific case proof of our theorem.

**Lemma G.1.** *(Allman et al., 2009) Consider a bivariate distribution of the form*

$$\mathbb{P}(X_1, X_2) = \sum_{i=1}^{r} \pi_i \mu_i^1(X_1) \mu_i^2(X_2). \tag{6}$$

*Then $\mathbb{P}(X_1, X_2)$ has rank $r$, if and only if, for each of $j = 1, 2$ the measures $\{\mu_i^j\}_{1 \leq i \leq r}$ are linearly independent, where $\mu_i^j$ denote the jth marginal of $\mu_i$.*

*Remark* G.2. One can see that, given $\mathbb{P}(X_1, X_2|\mathbf{V})$ has the same forms of Eq. 6, the above result still holds. Besides, Let $\mu_i^j$ represent a probabilistic decomposition of graphical models, $\mathbb{P}(X_j|V = i)$, with $|\operatorname{supp}(V)| = r$, under the Markov assumption and faithfulness condition, the above result also holds.

*Remark* G.3 (Mathematical Representation of Probability Tensor for Continuous Variables). To represent the probability tensor of continuous variable, such as $X_i, X_j$, one can let the entry of probability tensor $p_{i,j} \approx \int_{x_i}^{x_i+\Delta} \int_{x_j}^{x_j+\Delta} p(x_i, x_j) dx_i dx_j$, where $\Delta \to 0$, such that the probability tensor of $\mathbb{P}(X_i, X_j)$ has size $I \times J$, with $I, J \to \infty$. One can see that this may result in an infinite tensor, leading to the curse of dimensionality. We will provide a practical solution for estimation rank in Sec .4.

---

[4]https://academic.oup.com/jrsssb/article/85/3/684/7094061

**Lemma G.4** (Tensor Rank Condition in Two Variables Case). *Let $X_1, X_2$ be two variables in the nonlinear causal model with discrete latent confounders, suppose the Markov assumption, faithfulness assumption, and completeness condition hold. Then $\mathbb{P}(X_1, X_2)$ has rank $r$ ($r \ll n$) if and only if there exist a discrete variable set $\mathbf{S}$ with $|\operatorname{supp}(\mathbf{S})| = r$, that d-separates $X_1$ from $X_2$, and there are no other conditional set $\tilde{\mathbf{S}}$ that satifies $|\operatorname{supp}(\tilde{\mathbf{S}})| < r$.*

*Proof.* 'If' part: If $\mathbf{S}$ with $|\operatorname{supp}(\mathbf{S})| = r$, are the minimal conditional set that $d$-separates $X_1$ from $X_2$, one can rewrrite $\mathbb{P}(X_1, X_2) = \sum_{i=1}^{r} \mathbb{P}(X_1|\mathbf{S} = i)\mathbb{P}(X_2|\mathbf{S} = i)\mathbb{P}(\mathbf{S} = i)$ according to the Markov assumption. Denote $f_{X_j|\mathbf{S}=i}$ be $\mathbb{P}(X_j|\mathbf{S} = i)$, by completeness condition, one can see that $\{f_{X_j|\mathbf{S}=i}\}_{i=1}^{r}$ are linearly independent. This is because $\mathbb{E}[g(L)|x] = 0$ implies that $\{f_{L=i|X}\}_{i=1}^{r}$ are linearly independent. By the Bayes' theorem, $\{f_{X_j|\mathbf{S}=i}\}_{i=1}^{r}$ also are linearly independent (here $\mathbf{S}$ are discrete latent variables set due to the model assumption).

Conversely, due to $\{f_{X_j|\mathbf{S}=i}\}_{i=1}^{r}$ with $j = 1, 2$ are linear independent, the corresponding set of c.d.f.s $\{F_{X_j|\mathbf{S}=i}\}_{i=1}^{r}$ are also linear independent. One may choose collections of points $\{t_k^j\}_{k=1}^{r}$ such that the $r \times r$ matrices $M_k$ whose $i, j$-entries are $F_{X_j|\mathbf{S}=i}(t_k^j)$ have full rank. Then with $F$ denoting the c.d.f. for $\mathbb{P}$, the matrix $N$ with entries $F(t_i^1, t_j^2)$ can be expressed as

$$N = M_1^{\mathsf{T}} \operatorname{diag}(\mathbb{P}(\mathbf{S}))M_2, \tag{7}$$

and therefore has full rank. Otherwise, a similar factorization arising from the expression of $\mathbb{P}$ using fewer than $r$ summands shows that $N$ has rank smaller than $r$, which violates the completeness condition. Thus the rank of $\mathbb{P}$ is at least $r$, and since the faithfulness assumption, it has rank exactly $r$.

'Only if' part: If $X_1$ and $X_2$ are $d$-separated by a continuous variable set $\mathbf{X}_p$, then $r \to n$. This is because $\mathbb{P}(X_1, X_2) = \int f(X_1, X_2|\mathbf{X}_p)f(\mathbf{X}_p)d\mathbf{X}_p$ for any data point. It means that for any $r$, the $r \times r$ matrices $M_k$ whose $i, j$-entries are $F_{X_j|\mathbf{X}_p}$ have full rank. Thus, there does not exist a low-rank expression of $\mathbb{P}$ with $r \ll n$. Otherwise, $\mathbf{X}_p$ degenerates into a discrete distribution that violates the faithfulness assumption. This also holds for the case that the conditional set contains a continuous variable set.

Suppose $\mathbf{P}(X_1, X_2)$ has rank $r$ with $r \ll n$, then there exist a discrete variable set $\mathbf{S}$ that $d$-separates $X_1$ and $X_2$. Otherwise, the rank $r$ must close to $n$ (for example, $\mathbb{P}(X_1, X_2) = \mathbb{P}(X_1|X_2)\mathbb{P}(X_2)$ has large rank $r$ due to $X_2$ are a continuous conditional set). By the Markov assumption, $\mathbf{P}(X_1, X_2)$ has the form

$$\mathbb{P}(X_1, X_2) = \sum_{i=1}^{r} \mathbb{P}(\mathbf{S} = i)f(X_1|\mathbf{S} = i)f(X_2|\mathbf{S} = i). \tag{8}$$

According to Lemma G.1, $\{f(X_j|\mathbf{S} = i)\}_{1 \le i \le r}$, $j = 1, 2$, must be linear independent. If there exists a conditional set $\tilde{\mathbf{S}}$ with $|\operatorname{supp}(\tilde{\mathbf{S}})| < r$, it violate the linear independent of $\{f(X_j|\mathbf{S} = i)\}_{1 \le i \le r}$, $j = 1, 2$. Thus, $\mathbf{S}$ is the minimal conditional set with smallest cardinality of latent support.

Besides, for any no conditional set $\mathbf{S}_1$ with $|\operatorname{supp}(\tilde{\mathbf{S}}_1)| = r$, $\mathbb{P}(X_1, X_2|\tilde{\mathbf{S}}_1 = i)$ is not a rank-one matrix, therefore, it cannot be expressed by Eq. 8 according to the Markov assumption. If there exist $\tilde{\mathbf{S}}_1$ with $|\operatorname{supp}(\tilde{\mathbf{S}}_1)| = r$ that is constructed from the parameter space, such that $\mathbb{P}(X_1, X_2|\tilde{\mathbf{S}}_1 = i)$ is a rank-one matrix, it will violate the faithfulness assumption.

In summary, if $\mathbb{P}(X_1, X_2)$ has rank $r$ with $r \ll n$, then there exists conditional set $\mathbf{S}$ with smallest cardinarlity latent support that $d$-separates $X_1$ from $X_2$.

$\square$

**Corollary G.5.** *Under the Markov assumption, faithfulness assumption, if the conditional set for $X_1$ and $X_2$ contains a continuous variable set $\mathbf{V}_p$, then for $\mathbb{P}(X_1, X_2|\mathbf{V}_p)$, the result of Lemma G.4 also holds.*

*Proof.* The proof is straightforward. In the nonlinear causal model with latent confounders, all continuous variables are observational. Therefore, given $\mathbb{P}(X_1, X_2|\mathbf{V}_p)$, if the result of Lemma G.4 does not hold, then it violates the Markov assumption and faithfulness assumption. $\square$

Now, we can prove Theorem 3.4.

*Proof.* When $|\mathbf{X}_p| = 2$, the result holds according to Lemma G.4 and Corollary G.5. Next, we consider the case that $|\mathbf{X}_p| > 2$ and $\mathbf{X}_q = \emptyset$.

'If' part: if $\mathbf{S}$ $d$-separates any pair variables in $\mathbf{X}_p$, according to the Markov assumption, we have

$$\mathbb{P}(\mathbf{X}_p) = \sum_{i=1}^{r} \mathbb{P}(\mathbf{S} = i) \prod_{j=1}^{n} \mathbb{P}(X_j | \mathbf{S} = i), \tag{9}$$

where $\mathbb{P}(X_j | \mathbf{S} = i) = f_{X_j | \mathbf{S}}$, $\{f_{X_j | \mathbf{S}}\}_{i=1}^{r}$ are linear independent by the completeness condition. We further have $\{F_{X_j | \mathbf{S}}\}_{i=1}^{r}$ also linearly independent, such that the probability tensor can be expressed with entries $F(t_i^1, \cdots, t_r^n)$ (see a specific case in the proof of Lemma G.4). Furthermore, if there does not exist $\tilde{\mathbf{V}}_p$ with $|\operatorname{supp}(\tilde{\mathbf{V}}_p)| < r$, by the definition of tensor rank and the faithfulness assumption, $\mathbb{P}(\mathbf{X}_p)$ has rank $r$.

'Only if' part: if the probability tensor $\mathbb{P}(\mathbf{X}_p)$ has rank $r$, then there does not exist $X_i, X_j \in \mathbf{X}_p$, $X_i$ and $X_j$ are $d$-separated by a continuous variable set. Otherwise, the sub-tensor with entries $X_i, X_j$ has rank $r$ that is close to $n$ (See the proof of Lemma G.4). Thus, by the definition of tensor rank, let $\mathbb{P}(\mathbf{S} = i)$ is a constant and $\mathbb{P}(\mathbf{X}_p | \mathbf{S} = i)$ is a rank-one tensor, one has

$$\mathbb{P}(\mathbf{X}_p) = \sum_{i=1}^{r} \mathbb{P}(\mathbf{S} = i) \mathbb{P}(X_1 | \mathbf{S} = i) \otimes \cdots \otimes \mathbb{P}(X_n | \mathbf{S} = i), \tag{10}$$

where any pair of $\mathbb{P}(X_i | \mathbf{S} = i)$ and $\mathbb{P}(X_j | \mathbf{S} = i)$ are linearly independent by the completeness condition. We aim to prove that $\mathbf{S}$ is a discrete conditional set with the smallest cardinality of support that $d$-separates all variables in $\mathbf{X}_p$. There are three cases:

(1) If $\mathbb{P}(\mathbf{S})$ with $|\operatorname{supp}(\mathbf{S})| = r$ is arbitrarily constructed from the parameter space (i.e., $\mathbf{S}$ is not a node set in the causal graph $\mathcal{G}$), it will violate the faithfulness assumption.

(2) If $\mathbf{S}$ is not a conditional set that $d$-separates any pair variables in $\mathbf{X}_p$, without loss of generality, suppose there exists $X_i, X_j \in \mathbf{X}_p$, $X_i \not\perp\!\!\!\perp X_j | \mathbf{S}$, one can see that $\mathbb{P}(\mathbf{X}_p | \mathbf{S} = i)$ is not a rank-one tensor since the sub-tensor $\mathbb{P}(X_i, X_j, \mathbf{X}_p^{\backslash(i,j)} = \mathbf{c})$ is not a rank-one matrix according to the Markov assumption and faithfulness assumption (Hackbusch, 2012), where $\mathbf{X}_p^{\backslash(i,j)}$ denote $\mathbf{X}_p \setminus \{X_i, X_j\}$, and $\mathbf{X}_p^{\backslash(i,j)} = \mathbf{c}$ means that the values of $\mathbf{X}_p^{\backslash(i,j)}$ is fixed to $\mathbf{c}$.

(3) If $\mathbf{S}$ is a conditional set for $\mathbf{X}_p$ but not a smallest cardinality of support, without loss of generality, let $\tilde{\mathbf{S}}$ is the conditional set with smallest cardinality of support, denoted by $|\operatorname{supp}(\tilde{\mathbf{S}})| = k$, $k < r$, one has

$$\begin{aligned}
\mathbf{P}(\mathbf{X}_p) &= \sum_{i=1}^{k} \mathbb{P}(\tilde{\mathbf{S}} = i) \mathbb{P}(X_1 | \tilde{\mathbf{S}} = i) \otimes \cdots \mathbb{P}(X_n | \tilde{\mathbf{S}} = i) \\
&= \sum_{i=1}^{k} \tilde{\mathbb{P}}(X_1 | \tilde{\mathbf{S}} = i) \otimes \cdots \mathbb{P}(X_n | \tilde{\mathbf{S}} = i) \\
&= \sum_{i=1}^{k} \mu_1^i \otimes \cdots \otimes \mu_n^i,
\end{aligned} \tag{11}$$

where $\tilde{\mathbb{P}}(X_1 | \tilde{\mathbf{S}} = i)$ denote $\mathbb{P}(\tilde{\mathbf{S}} = i) \mathbb{P}(X_1 | \tilde{\mathbf{S}} = i)$, and $\mu_j^i$ is a vector. According to the definition of tensor rank, the rank of $\mathbf{P}(\mathbf{X}_p)$ is $k$, which violates the preconditions that $\mathbf{P}(\mathbf{X}_p)$ has rank $r$. Thus, $\mathbf{S}$ must be the conditional set with the smallest cardinality of support.

Next, consider the case of $\mathbf{X}_q \neq \emptyset$, if $\mathbf{X}_q \cup \mathbf{S}$ $d$-separates any pair variable in $\mathbf{X}_p$, by the Markov assumption and completeness condition, one can see that $\mathbb{P}(\mathbf{X}_p | \mathbf{X}_q)$ has rank $r$. Without considering the $\mathbf{X}_q$ as the conditional set, the rank of $\mathbb{P}(\mathbf{X}_p)$ will be close to $n$. This is because there exists $X_i, X_j \in \mathbf{X}_p$, $X_i, X_j$ are $d$-connected without given $\mathbf{X}_q$ as the conditional set. Thus, the sub-tensor with entries $X_i$ and $X_j$ has a rank $k$ close to $n$ (see the proof in Lemma G.4).

On the other hand, suppose $\mathbb{P}(\mathbf{X}_p | \mathbf{X}_q)$ has rank $r$, and $r \ll n$, according to the previous proof, there exists a conditional set $\mathbf{S}$ with the smallest cardinality of support $r$ that $d$-separates variable in $\mathbf{P}(X_p | \mathbf{X}_q)$. Since $\mathbf{P}(X_p | \mathbf{X}_q)$ are the conditional

distribution of $\mathbf{X}_p$ given $\mathbf{X}_q$, then $\mathbf{X}_q$ also be the conditional set of $\mathbf{X_p}$.

$\square$

**Corollary G.6.** *If $\mathbf{X}_q$ is the conditional set that d-separates all variables in $\mathbf{X}_p$, then the conditional probability tensor $\mathbb{P}(\mathbf{X}_p|\mathbf{X}_q)$ has rank one.*

*Proof.* Since $\mathbf{X}_q$ is the conditional set, we have

$$\mathbb{P}(\mathbf{X}_p|\mathbf{X}_q) = f_{X_1|\mathbf{X}_q} \cdots f_{X_n|\mathbf{X}_q} = \mathbb{P}(X_1|\mathbf{X}_q) \otimes \cdots \otimes \mathbb{P}(X_n|\mathbf{X}_q) \tag{12}$$

is a rank-one tensor according to the Markov assumption. Thus, the conditional probability tensor has rank one.

$\square$

### G.2. Proof of Proposition 4.2

*Proof.* We will show that, there exists a partition $\{\mathcal{X}\}_{i=1}^I$ with $I \geq r$ such that the matrix $\mathbb{P}(\tilde{X}|L)$ has rank $r$. We first show that the function $\{f_{L|X}(L = l|x)\}_{l=1}^r$ are linear independent. Prove by contradiction. Suppose that there are $r$ numbers $\{\alpha_l\}_{l=1}^r$ that are not all zero such that $\sum_{l=1}^r \alpha_l f_{L|X}(L = l|x) = 0$. Let $g(l)$ be the piece-wise constant function defined by $g(l) = \alpha_l$, we then have $\sum_{l=1}^r g(l) f_{L|X}(L = l|x) = 0$ for all $x$, i.e., $\mathbb{E}[g(l)|x] = 0$. Since $g(\cdot) \neq 0$, this contradicts the completeness assumption. Consequently, $\{f_{L|X}(L = l|x)\}_{l=1}^r$ are linear independent. Besides, note that $f_{X|L}(x|L = l) = \frac{f_{L|X}(L=l|x)f_X(x)}{\mathbb{P}(L=l)}$, therefore it is easy to obtain the linear independence of $\{f_{X|L}(x|L = l)\}_{l=1}^r$. Since $f_{X|L}(x|L = l) = F'_{X|L}(x|L = l)$, we can also obtain the linear independence of $\{F_{X|L}(x|L = l)\}_{l=1}^r$ (Similar results can be found in (Liu et al., 2024)).

Given that $\{F_{X|L}(x|L = l)\}_{l=1}^r$ are linear independent, we now show that there exists $I \geq r$ points $\mathbf{x}_I = \{x_i\}_{i=1}^I$ such that the matrix $\mathbb{P}(\tilde{X}|L)$ has rank $r$. Denote $F_{X|l}(\mathbf{x}_I) = [F_{X|l}(x_1), \cdots, F_{X|l}(x_I)]^\intercal$, where $F_{X|l}(x) = F_{X|L}(x|L = l)$. We have

$$\text{Rank}(\mathbb{P}(\tilde{X}|L)) = \text{Rank}\left(\begin{bmatrix} F_{X|1}(x_1) & \cdots & F_{X|r}(x_1) \\ F_{X|1}(x_2) & \cdots & F_{X|r}(x_2) \\ \cdots & \cdots & \cdots \\ F_{X|1}(x_I) & \cdots & F_{X|r}(x_I) \end{bmatrix}\right), \tag{13}$$

i.e., $\text{Rank}([F_{X|1}(\mathbf{x}_I), \cdots, F_{X|r}(\mathbf{x}_I)]) = r$.

We first discuss the case when $I = r$. The proof is based on mathematical induction:

(1) Base: $\forall l \in [r]$, we have $\exists x_l, F_{X|l}(x_l) \neq 0$. Thus, we have $\det(F_{X|l}(x_l)) \neq 0$.

(2) Induction hypothesis: assume for any combination $l_1, l_2, \cdots, l_k \in [r]$ ($k \in \mathbb{N}, 1 \leq k \leq r - 1$), for $k$ linear independent functions $F_{X|l_1}(\cdot), \cdots F_{X|l_k}(\cdot)$, there exists $k$ points $\mathbf{x}_k = \{x_1, \cdots, x_k\}$ such that $\det(M_k) \neq 0$, where $M_k \triangleq [F_{X|l_1}(\mathbf{x}_k), \cdots, F_{X|l_k}(\mathbf{x}_k)]$.

(3) Induction: show that for any $k + 1$ linear independent functions $F_{X|l_1}(\cdot), \cdots, F_{X|l_{k+1}}(\cdot)$ ($k \in \mathbb{N}, 2 \leq k + 1 \leq r$), there exists $k + 1$ point $\mathbf{x}_{k+1} = \{x_1, \cdots, x_{k+1}\}$ such that $\det(M_{k+1}) \neq 0$, where $M_{k+1} \triangleq [F_{X|l_1}(\mathbf{x}_{k+1}), \cdots, F_{X|l_{k+1}}(\mathbf{x}_{k+1})]$. To this end, we first have the Laplace expansion of matrix determinant

$$\det(M_{k+1})(\mathbf{x}_{k+1}) = \sum_{l=1}^{k+1}(-1)^{k+1+l} \det(A_l^{k+1})F_{X|l}(x_{k+1}), \tag{14}$$

where $A_l^{k+1}$ denote the submatrix obtained by removing the $(k + 1)$-th row and $l$-th column from $M_{k+1}$. That is, $A_l^{k+1} = [F_{X|l_1}(\mathbf{x}_k), \cdots, F_{X|l_{l-1}}(\mathbf{x}_k), F_{X|l_{l+1}}(\mathbf{x}_k), \cdots, F_{X|l_{k+1}}(\mathbf{x}_k)]$ with $F_{X|l}(\mathbf{x}_k) = [F_{X|l}(x_1), \cdots, F_{X|l}(x_k)]^\intercal$.

Since $F_{X|l_1}(\cdot), \cdots, F_{X|l_k}(\cdot)$ are $k$ linear independent functions, according to the induction hypothesis, there exist $k$ points $x_1, \cdots, x_k$ such that $\det(A_l^{k+1}) \neq 0$. That is, at least one of the cofactors $\det(A_l^{k+1})$ in the expansion is not zero. Since $F_{X|l_1}(\cdot), \cdots F_{X|l_{k+1}}(\cdot)$ are linear independent, we have $\sum_{l=1}^{k+1} \alpha_l F_{X|l} = 0$ iif. all $\alpha_l = 0$ (a column expansion of matrix determinant). Thus, there exists $x_{k+1}$ such that $\det(M_{k+1})(\mathbf{x}_{k+1}) \neq 0$ for $\mathbf{x}_{k+1}$, which concludes the induction.

For the case when $I > r$, we have $F_{X|1}(\mathbf{x}_I), \cdots, F_{X|r}(\mathbf{x}_I)$ (where $F_{X|l}(\mathbf{x}) = [F_{X|l}(x_1), \cdots, F_{X|l}(x_I)]^\intercal$) are linear independent because $F_{X|1}(\mathbf{x}_r), \cdots, F_{X|r}(\mathbf{x}_r)$ (where $F_{X|l}(\mathbf{x}_r) = [F_{X|l}(x_1), \cdots, F_{X|l}(x_r)]^\intercal$) are linear independent. That is, the row slice of matrix $M_r \triangleq [F_{X|1}(\mathbf{x}_I), \cdots, F_{X|r}(\mathbf{x}_I)]$ has the full column rank and hence the $M_r$ also has full column rank. $\qquad\square$

## G.3. Proof of Lemma 4.4

*Proof.* Any discretization of $X_i$ and $X_j$ in the discrete latent variable model has an upper bound of $|\operatorname{supp}(L)|$, in which $L$ $d$-separates $X_i$ from $X_j$. For convenience, we denote $\Delta_i$ as the support of $X_i$ that $\tilde{X}_i = i$. $\Delta_j$ is defined similarly.

$$
\begin{aligned}
\mathbb{P}(\tilde{X}_i = i, \tilde{X}_j = j) &= \int_{\Delta_i} \int_{\Delta_j} \mathbb{P}(X_i, X_j) dX_i dX_j \\
&= \int_{\Delta_i} \int_{\Delta_j} \sum_{k=0}^{\operatorname{supp}(L)} \mathbb{P}(X_i, X_j | L = k)\mathbb{P}(L = k) dX_i dX_j \\
&= \sum_{k=0}^{\operatorname{supp}(L)} \mathbb{P}(L = k) \int_{\Delta_i} \int_{\Delta_j} \mathbb{P}(X_i | L = k)\mathbb{P}(X_j | L = k) dX_i dX_j \\
&= \sum_{k=0}^{\operatorname{supp}(L)} \mathbb{P}(L = k) \int_{\Delta_i} \mathbb{P}(X_i | L = k) dX_i \int_{\Delta_j} \mathbb{P}(X_j | L = k) dX_j \\
&= \sum_{k=0}^{\operatorname{supp}(L)} \mathbb{P}(L = k)\mathbb{P}(\tilde{X}_i = i | L = k)\mathbb{P}(\tilde{X}_j = j | L = k),
\end{aligned}
\tag{15}
$$

where $\mathbb{P}(\tilde{X}_i = i | L = k) = \int_{\Delta_i} \mathbb{P}(X_i | L = k) dX_i$ and $\mathbb{P}(\tilde{X}_j = j | L = k) = \int_{\Delta_j} \mathbb{P}(X_j | L = k) dX_j$.

We can rewrite $\mathbb{P}(\tilde{X}_i = i, \tilde{X}_j = j)$ as follows.

$$
\begin{aligned}
&\sum_{k=0}^{\operatorname{supp}(L)} \mathbb{P}(L = k)\mathbb{P}(\tilde{X}_i = i | L = k)\mathbb{P}(\tilde{X}_j = j | L = k) \\
&= \mathbb{P}(\tilde{X}_i = i | L) \operatorname{diag}(\mathbb{P}(L))\mathbb{P}^\intercal(\tilde{X}_j = j | L).
\end{aligned}
\tag{16}
$$

So, one can get that

$$
\mathbb{P}(\tilde{X}_i, \tilde{X}_j) = \mathbb{P}(\tilde{X}_i | L) \operatorname{diag}(\mathbb{P}(L))\mathbb{P}^\intercal(\tilde{X}_j | L).
\tag{17}
$$

Thus, if $|\operatorname{supp}(L)| < |\operatorname{supp}(\tilde{X}_i)|$, and $\operatorname{supp}(L)| < |\operatorname{supp}(\tilde{X}_j)|$, the upper bound of rank of $\mathbb{P}(\tilde{X}_i, \tilde{X}_j)$ is

$$
\operatorname{Rank}(\mathbb{P}(\tilde{X}_i, \tilde{X}_j)) \le |\operatorname{supp}(L)|.
\tag{18}
$$

Moreover, one can see that $\operatorname{Rank}(\mathbb{P}(\tilde{X}_i, \tilde{X}_j)) = |\operatorname{supp}(L)|$ if and only if $\mathbb{P}(\tilde{X}_i | L)$ and $\mathbb{P}(\tilde{X}_j | L)$ are full column rank.

$\qquad\square$

## G.4. Proof of Theorem 4.5

*Proof.* This result is proven based on Lemma 4.4. Since $\operatorname{Rank}(\mathbb{P}^{(k)}(\tilde{X}_i, \tilde{X}_j)) \le |\operatorname{supp}(L)|$, and the upper bound is achieved if $\mathbb{P}(\tilde{X}_i | L)$ and $\mathbb{P}(\tilde{X}_j | L)$ are full column rank, we cannot find a discretization such that $\operatorname{Rank}(\mathbb{P}^{(k)}(\tilde{X}_i, \tilde{X}_j)) > |\operatorname{supp}(L)|$, i.e., for the $(k+1)$-th time discretization, the rank stops increasing. Therefore, in this case, $\tilde{X}_i$ and $\tilde{X}_j$ satisfy the full-rank assumption. $\qquad\square$

## G.5. Proof of Proposition 4.7

*Proof.* In the discrete latent structure model, any two observed variables $X_i, X_j$ are conditionally independent given one latent parent of them. We first show that, by discretizing the marginal distribution of $X_i$ or $X_j$, the generation process of $\tilde{X}$ is still determined by its latent parent. One can see that

$$
\begin{aligned}
\mathbb{P}(\tilde{X} = i) &= \int_{B_{i-1}}^{B_i} P(X)dX \\
&= \int_{B_{i-1}}^{B_i} \sum_{j=1}^{|\mathrm{supp}(\mathbb{P}(L))|} \mathbb{P}(X|L=j)\mathbb{P}(L=j)dX \\
&= \sum_{j=1}^{|\mathrm{supp}(\mathbb{P}(L))|} \int_{B_{i-1}}^{B_i} \mathbb{P}(X|L=j)\mathbb{P}(L=j)dX \\
&= \sum_{j=1}^{|\mathrm{supp}(\mathbb{P}(L))|} \mathbb{P}(L=j) \int_{B_{i-1}}^{B_i} \mathbb{P}(X|L=j)dX \\
&= \sum_{j=1}^{|\mathrm{supp}(\mathbb{P}(L))|} \mathbb{P}(L=j)\mathbb{P}(\tilde{X}=i|L=j) \\
&= \mathbb{P}(L)\mathbb{P}^{\mathsf{T}}(\tilde{X}=i|L)
\end{aligned}
\tag{19}
$$

Under Markov assumption, faithfulness assumption and $\mathbb{P}(\tilde{X}|L)$ is full rank, we have

$$
\mathbb{P}(\tilde{X}) = \mathbb{P}(L)\mathbb{P}(\tilde{X}|L),
\tag{20}
$$

which means that the generation process of $X$ is still determined by its latent parent $L$. Thus, we have

$$
\mathbb{P}(\tilde{X}_1) = \mathbb{P}(L_1)\mathbb{P}(\tilde{X}_1|L_1), \mathbb{P}(\tilde{X}_2) = \mathbb{P}(L_2)\mathbb{P}(\tilde{X}_2|L_2),
\tag{21}
$$

where $L_1$ is parent of $X_1$ and $L_2$ is parent of $X_2$.

Suppose $L_1 \perp\!\!\!\perp L_2|L$ in the causal graph, we aim to show that $\tilde{X}_1 \perp\!\!\!\perp \tilde{X}_2|L$ also holds.

$$
\begin{aligned}
&\mathbb{P}(\tilde{X}_1, \tilde{X}_2, L_1, L_2, L) \\
&= \mathbb{P}(\tilde{X}_1|L_1)\mathbb{P}(\tilde{X}_2|L_2)\mathbb{P}(L_1|L)\mathbb{P}(L_2|L)\mathbb{P}(L) \\
&= \underbrace{\mathbb{P}(\tilde{X}_1|L_1)\mathbb{P}(L_1|L)}_{\mathbb{P}(\tilde{X}_1, L_1|L)} \underbrace{\mathbb{P}(\tilde{X}_2|L_2)\mathbb{P}(L_2|L)}_{\mathbb{P}(\tilde{X}_2, L_2|L)} \mathbb{P}(L).
\end{aligned}
\tag{22}
$$

Moreover, let $\mathbb{P}(\tilde{X}_1|L) = \sum_{L_1} \mathbb{P}(\tilde{X}_1, L_1|L)$, and $\mathbb{P}(\tilde{X}_2|L) = \sum_{L_2} \mathbb{P}(\tilde{X}_2, L_2|L)$, one can see that

$$
\mathbb{P}(\tilde{X}_1 \tilde{X}_2) = \sum_{L} \mathbb{P}(\tilde{X}_1|L)\mathbb{P}(\tilde{X}_2|L)\mathbb{P}(L).
\tag{23}
$$

Therefore, $\tilde{X}_1 \perp\!\!\!\perp \tilde{X}_2|L$ also holds.

Meanwhile, if $L_1 \not\!\perp\!\!\!\perp L_2|L$ in the causal graph, we aim to show that $\tilde{X}_1 \not\!\perp\!\!\!\perp \tilde{X}_2|L$ also holds. Without loss of generality, suppose $\mathbb{P}(L_1 L_2|L) \neq \mathbb{P}(L_1|L)\mathbb{P}(L_1|L)$, one can see that

$$\mathbb{P}(\tilde{X}_1, \tilde{X}_2, |L = i)$$
$$= \sum_{L_1} \sum_{L_2} \mathbb{P}(\tilde{X}_1|L_1)\mathbb{P}(\tilde{X}_2|L_2)\mathbb{P}(L_1, L_2|L = i)$$
$$= \sum_{L_1} \mathbb{P}(\tilde{X}_1|L_1) \sum_{L_2} \mathbb{P}(\tilde{X}_2|L_2)\mathbb{P}(L_1, L_2|L = i) \tag{24}$$
$$\neq \mathbb{P}(\tilde{X}_1|L = i)\mathbb{P}(\tilde{X}_2|L = i).$$

Thus, we have $\tilde{X}_1 \not\perp\!\!\!\perp \tilde{X}_2|L$ hold if $X_1 \not\perp\!\!\!\perp X_2|L$ in the causal graph. Based on the proof above, the CI relations after discretization are consistent. $\qquad\square$

**Discussion.** Actually, since the CI relationship between observed variables relies on the CI relations among latent variables in the mixed-type latent structure model (LSM), the discretized variables still maintain the conditional probability decomposition in the discrete probability space. In other words, the discretization does not alter the relations within the discrete probability space.

### G.6. Proof of Lemma 4.8

*Proof.* For continuous observational data that satisfy the completeness condition, one can discretize the continuous data into discretized data that satisfy the non-degeneracy condition by Theorem 4.5. Thus, for the discretized data $\tilde{\mathbf{X}}_p$, it satisfies the Markov assumption, faithfulness assumption, and non-degeneracy condition according to Proposition 4.7, the tensor rank condition holds for $\tilde{\mathbf{X}}_p$. The proof for the discrete data can be referred to (Chen et al., 2024). $\qquad\square$

### G.7. Proof of Proposition 4.11

*Proof.* Under the assumption that each latent variable has the same cardinality of support, it is easy to check if $X_i$ is only caused by one latent parent. One can prove it by contradiction. If $X_i$ has more than one latent parent, by the model assumption, there exists $X_j$ that shares the set of common latent parents with $X_i$, thus, $\mathrm{Rank}(\mathbb{P}(X_i, X_j)) > r$ according to the graphical implication of the tensor rank condition, where $r$ is the cardinality of latent support. This means that if $X_i$ is not a one-factor structure, there are $\mathrm{Rank}(\mathbb{P}(X_i, X_j)) > r$ and $\mathrm{Rank}(\mathbb{P}(X_i, X_k)) = r$ where $X_k$ is any one-factor structure variable. Thus, $\mathrm{Rank}(\mathbb{P}(X_i, X_j))$ is different for any $X_j \in \mathbf{X} \setminus \{X_i\}$.

$\qquad\square$

### G.8. Proof of Proposition 4.13

*Proof.* Due to $\mathbf{X}_c$ being a one-factor group, $\{X_i, X_j\}$ also be a one-factor sub-group. If $\{X_i, X_j, X_k\}$ is not a one-factor cluster, without loss of generality, let $L_1$ be the parent of $\{X_i, X_j\}$ and $L_2$ be the parent of $X_k$, according to the model assumption, there exist $X_s$ also be the children of $L_2$, such that $\{X_i, X_j, X_k, X_s\}$ are $d$-separated by $\{L_1, L_2\}$. By the graphical implication of the tensor rank condition, $\mathrm{Rank}(\mathbb{P}(X_i, X_j, X_k, X_s)) = r^2$ (The similar proof can be seen in (Chen et al., 2024)). Thus, $\{X_i, X_j, X_k\}$ is a causal cluster if $\forall X_s \in \mathbf{X} \setminus \{X_i, X_j, X_k\}$, $\mathrm{Rank}(\mathbb{P}(X_i, X_j, X_k, X_s)) = r$. $\qquad\square$

### G.9. Proof of Proposition 4.15

*Proof.* By Proposition 4.11, the observed variable can be classified into the one-factor group and the multi-factor group. That is, $\mathbf{X} \setminus \mathbf{X}_c$ is the set of the multi-factor group. For $\mathbf{C} = \{X_i, X_j, X_k\}$, let $\{L_1, \cdots, L_n\}$ be the latent parent set of $\mathbf{C}$, one can map $\{L_1, \cdots, L_n\}$ to be one latent variable $L_q$ with cardinality $r^n$. Then this result can be proved by the extension result of different state space cases in (Chen et al., 2024). $\qquad\square$

### G.10. Proof of Proposition 4.16

*Proof.* For the first proposition, because $\mathbf{C}_1 \cap \mathbf{C}_2 \neq \emptyset$, with loss of generality, assume that $\mathbf{C}_1 \cap \mathbf{C}_2 = X_k$. Let $Pa(X_k)$ denote the parent of $X_k$. One has $Pa(X_k)$ be the parent of $\mathbf{C}_1$ according to the definition of the causal cluster. Meanwhile, we also have $Pa(X_k)$ be the parent of $\mathbf{C}_2$. Therefore, $\mathbf{C}_1$ and $\mathbf{C}_2$ share a common latent parent.

For the second proposition, since $|Pa(\mathbf{C}_1)| > |Pa(\mathbf{C}_2)|$, and $\mathrm{Rank}(\mathbb{P}(\mathbf{C}_1 \cup \mathbf{C}_2)) = \mathrm{Rank}(\mathbb{P}(\mathbf{C}_1))$, by the graphical

implication of tensor rank, one can infer $Pa(\mathbf{C}_1)$ also be the $d$-separated set for $\mathbf{C}_2$. Under the model assumption of discrete mixed-LSM, one can infer that $Pa(\mathbf{C}_2) \subseteq Pa(\mathbf{C}_1)$, and hence $\mathbf{C}_1$ and $\mathbf{C}_2$ share the common latent parent. □

### G.11. Proof of Theorem 4.17

*Proof.* Proof of this theorem is straightforward by combining the proof of (Chen et al., 2024) and the property of tensor rank in which the tensor rank is greater than or equal to the sub-tensor rank (constructed by sufficient measured variables). □

### G.12. Proof of Theorem 4.19

*Proof.* We prove the identification by a two-stage algorithm, including the identification of the measurement model and the identification of the structure model.

Stage I: the identification of the measurement model. In this stage, Algorithm 1 first discretizes all continuous variables to enable efficient testing of tensor rank. Next, it identifies the latent support and one-factor group using Proposition 4.13. By applying Propositions 4.13 and 4.15, the algorithm identifies the causal clusters and determines the number of latent parents for each cluster. Finally, according to Proposition 4.16, the causal clusters are merged to prevent the redundant introduction of latent variables. By this, the measurement model is completely identified under the Mixed LSM.

Stage II: Identification of the Structural Model. In this stage, $d$-separation relations are identified using Theorem 4.17. Then, the PC algorithm (Spirtes et al., 2000) is applied. Given the measurement model, the causal structure among latent variables can be identified up to a Markov equivalence class (Chen et al., 2024).

□

# H. More details on the Experimental Results

We first introduce a detailed definition of different evaluation metrics. Denote the output results of each algorithm $G_{out}$, the true graph is labeled $G$: The performance of the causal cluster is evaluated by following the scores:

**latent omission**: the number of latents in $G$ that do not appear in $G_{out}$ divided by the total number of true latents in $G$;

Moreover, we use the F1 score to evaluate the performance of causal structure among latent variables. Next, we describe the calculation details of F1 score. First, we introduce two metrics used basically for F1 score.

**edge omission (EO)**: the number of edges in the structural model of $G$ that do not appear in $G_{out}$ divided by the possible number of edge omissions;

**edge commission (EC)**: the number of edges in the structural model of $G_{out}$ that do not exist in $G$ divided by the possible number of edge commissions;

$$F1 = \frac{2 \times (1 - EO) \times (1 - EC)}{(1 - EO) + (1 - EC)}.$$

### H.1. Experimental Results on Real-world Dataset

In this section, we apply our algorithm to the real-world Industrialization and Political Democracy dataset (Bollen, 1989). This dataset contains various measures of political democracy and industrialization across developing countries, with 75 observations of 11 variables. The details of each observation are summarized as follows.

- y1: Expert ratings of the freedom of the press in 1960

- y2: The freedom of political opposition in 1960

- y3: The fairness of elections in 1960

- y4: The effectiveness of the elected legislature in 1960

- y5: Expert ratings of the freedom of the press in 1965

- y6: The freedom of political opposition in 1965

- y7: The fairness of elections in 1965

- y8: The effectiveness of the elected legislature in 1965

- x1: The gross national product (GNP) per capita in 1960

- x2: The inanimate energy consumption per capita in 1960

- x3: The percentage of the labor force in industry in 1960

In (Bollen, 1989), the authors show that there are three latent factors that cause these observed variables. Specifically, $\{x1, x2, x3\}$ are driven by a single latent factor, while $\{y1, y2, y3, y4\}$ and $\{y5, y6, y7, y8\}$ are each influenced by a two-factor structure. In our implementation, we first obtain the discretized data using our discretization technique, where each observed variable is discretized into a discrete variable with four categories. We then apply a bootstrapping resampling approach to enhance the statistical properties of the data. Next, we compute the results for all possible combinations of sets with four elements, sorting them according to their reconstruction error (CP decomposition with rank $r$, more details can refer to (Chen et al., 2024)) and selecting the top-$K$ items. (where $K = 10$). Finally, our algorithm outputs the following result:

- $\{L_1\} \to \{x1, x2, x3\}, |\operatorname{supp}(\mathbb{P}(L_1))| = 3$,

- $\{L_2, L_3\} \to \{y1, y2, y3, y4, y5, y6, y7, y8\}, |\operatorname{supp}(\mathbb{P}(L_2))| = 2, |\operatorname{supp}(\mathbb{P}(L_3))| = 2$.

One can observe that the results are consistent with the discussion in (Bollen, 1989), demonstrating the effectiveness of our algorithm, including the discretization technique. In fact, these latent variables have real-world meaning. For example, as discussed in (Bollen, 1989), $y1$ to $y4$ are intended to be indicators of the latent variable *political democracy in 1960*, $y5$ to $y8$ are indicators of *political democracy in 1965*, and $x1$ to $x3$ are indicators of *industrialization in 1960*. Since there are common features between political democracy in 1960 and political democracy in 1965, our algorithm learns them as a two-factor structure.

