# OpenReview forum: "Identification of Latent Confounders via Investigating the  Tensor Ranks of the Nonlinear Observations"
_ICML.cc/2025/Conference — ICML 2025 poster_

### Official Review · Reviewer_gVAe · 2025-03-13

**Overall Recommendation:** 4

**Summary:**

This paper studies the problem of learning discrete latent variable causal structures from mixed-type observational data using the graphical criteria of tensor rank conditions. To handle continuous observed variables, the author proposes a discretization method that ensures the discretized data satisfy the full-rank assumption, thereby allowing existing results for discrete data to be directly extended. The proposed method is further applied to the task of learning discrete latent variable causal structures from mixed-type observational data.

**Claims And Evidence:**

The claims are generally well-supported by theoretical analysis, but certain aspects, such as novelty and baseline comparisons, require further clarification.

**Essential References Not Discussed:**

The author gives a well-reviewed on the related literature.

**Experimental Designs Or Analyses:**

The experimental design and analyses are generally sound. However, the simulated structure appears relatively simple, which may limit the evaluation of the method’s robustness in more complex settings.

**Methods And Evaluation Criteria:**

The proposed method is reasonable and well-motivated. The evaluation metrics selected are appropriate for the problem of latent confounder identification.

**Other Comments Or Suggestions:**

1. Could the author clarify the key differences between the tensor rank condition in non-linear causal models and the tensor rank condition in discrete latent structure models? How does this work advance previous results, particularly in relation to [1]?

2. For the case of continuous observed variables and discrete latent variables, previous work—such as the Mixture Oracle method—has also explored identifiability. What are the key differences between your approach and the Mixture Oracle method?

Reference:

[1]. Learning Discrete Latent Variable Structures with Tensor Rank Conditions. NeurIPS 2024.

**Other Strengths And Weaknesses:**

Strengths:

1. The paper is well-written and clearly organized.

2. It provides a comprehensive review of related literature.

3. The problem of learning latent structures in non-linear models is both important and challenging. The proposed approach appears to be sound and well-justified.

Weaknesses:

1. The paper focuses on a specific class of non-linear models, where latent variables are discrete and observed variables are continuous, which may limit general applicability.

2. The tensor rank condition and its graphical criteria have been explored in prior work, so the novelty needs further clarification.

3. The Mixture Oracle method, a relevant baseline, is not included in the experimental comparison, which limits the evaluation.

**Questions For Authors:**

see above

**Relation To Broader Scientific Literature:**

The proposed method contributes to the problem of latent confounder identification. While the approach is interesting and relevant, its novelty and practical advantages over existing methods should be better articulated.

**Theoretical Claims:**

The theoretical claims are clearly presented, and the proofs are OK.

---

> ### Author Rebuttal · Authors · 2025-03-31
>
> Thanks for your careful and valuable comments. We will respond to these issues point by point.
> >[W] The Mixture Oracle method, a relevant baseline, is not included in the experimental comparison, which limits the evaluation.
>
> **WQ1**: We have added experimental results using the Mixture Oracle as a baseline. Since K-means is not suitable for discrete data, we adapt the Mixture Oracle implementation by replacing K-means with the K-modes algorithm [1]. We evaluate the accuracy in identifying the latent support and include the results in the updated experiments. **As shown in Table 1 of https://anonymous.4open.science/r/TensorRank-E052/Experimental%20results.pdf**
>
> One can observe that the Mixture Oracle method does not effectively identify the support of latent variables, even when using a discrete clustering algorithm. This may be due to the fact that clustering yields only an approximate solution and lacks theoretical guarantees for recovering the true latent structure.
> >[Q1] Could the author clarify the key differences between the tensor rank condition in non-linear causal models and the tensor rank condition in discrete latent structure models? How does this work advance previous results, particularly in relation to (Chen et al. 2024)?
>
> **Q1A1**: We would like to clarify our contributions in two main aspects—**applicability conditions and identification boundaries**—particularly in relation to the work by Chen et al. (2024).
>
> 1. **Our work considers a more general setting for when the tensor rank condition holds.** In particular, we allow the observed variables to be continuous and show that the latent structure remains identifiable when each latent variable has two sufficiently informative observed variables. In contrast, Chen et al. (2024) propose the tensor rank condition when all variables  are discrete and each observed variable is assumed to be a sufficient measurement for its latent parent (i.e., the observed support has higher cardinality than the latent’s).
>
> 2. **We also extend the tensor rank condition to conditional probability tables, which leads to a more general identifiability condition for latent structures**, as described in Appendix D. While Chen et al. (2024) explore rank constraints based on the joint probability table under the assumption of a three-pure-measurement-variable model, our approach considers constraints on conditional distributions. This allows us to test conditional relationships among observed variables in the presence of latent confounding (i.e., impure structures), and further, to identify the causal structure among latent variables even under impure setting (Appendix D). To the best of our knowledge, this provides a novel identifiability condition for discrete latent variable models.
>
>     To evaluate structure learning under an impure setting, we conduct the following simulation, **as shown in  Table 2 of https://anonymous.4open.science/r/TensorRank-E052/Experimental%20results.pdf**. We compare our method (Appendix D.2) with the approach proposed by Chen et al. (2024). One can see that the method by (Chen et al. 2024) cannot learn the causal structure among latent variable, this is because their method only suitable for the pure measurement model.
>
>     Moreover, in Appendix D.3, we show that only two purely measured variables per latent variable are sufficient to identify the measurement model when the latent structure is fully connected. **This result also relaxes the structural assumptions required by previous work based on the three-pure-measurement-variable assumption.**
> >[Q2] For the case of continuous observed variables and discrete latent variables, previous work—such as the Mixture Oracle method—has also explored identifiability. What are the key differences between your approach and the Mixture Oracle method?
>
> **Q2A2**: We would like to clarify our contributions with two main points, especially in relation to the Mixture Oracle approach.
>
> 1. **We offer a robust and testable method for structure learning**. Unlike the Mixture Oracle method, which relies on identifying a mixture model (as discussed in our Appendix E) and provides only an approximate method for parameter estimation—an approach that we find can be unreliable and difficult to test (see Sec. 3.1 and App. B)—our method introduces a hypothesis test for the tensor rank condition. This test not only enhances robustness compared to the Mixture Oracle but also allows us to identify the latent support with theoretical guarantees (see **W3A3** in response to Reviewer e9yn).
>
> 2. **We introduce a novel and more general structural condition for the identifiability of discrete latent structures** (refer to Point 2 in **Q1A1**). In contrast, the Mixture Oracle assumes that no edges between observed variables, which is a quite restrictive assumption.
>
> We sincerely appreciate your thoughtful inquiry and hope this clarification helps. Please feel free to reach out if you have any further questions.

---

> > ### Comment · Reviewer_gVAe · 2025-04-04
> >
> > Thank you for the author's response—The rebuttal addressed my concerns.
> > I believe this work makes a meaningful contribution to the existing literature, particularly in terms of applicability conditions and identification boundaries for tensor rank. At this point, I’m curious whether the identifiability bounds of tensor rank are well studied. If so, can this structure condition be extended to hierarchical settings, potentially enabling applications such as understanding image representations (see [1])?
> > I have raised my score to “Weak Accept.” I may consider increasing it further if a compelling real-world application scenario is provided.
> > Reference:
> > [1] Learning Discrete Concepts in Latent Hierarchical Models. NeurIPS 2024.

---

> > > ### Author Response · Authors · 2025-04-05
> > >
> > > Thank you for your positive assessment and constructive suggestions. Below, we address the two issues—on theoretical bounds and practical applications—point by point.
> > >
> > > > [**Question 1**] I’m curious whether the identifiability bounds of tensor rank are well studied. If so, can this structure condition be extended to hierarchical settings, potentially enabling applications such as understanding image representations (see [1])?
> > >
> > > **A1**: Thank you for the thoughtful question. **To our knowledge, the identifiability bounds of tensor rank have been extensively studied in our work, particularly in the context of joint distributions, subtensors, and conditional probability tensors.** Achieving stronger identifiability typically requires imposing additional structural assumptions or leveraging higher-dimensional tensor rank constraints, which tend to be more computationally expensive and statistically less stable in practice.
> > >
> > > For instance, to identify latent support, we rely only on two-dimensional tensor rank, which leads to the two-sufficient-measurement assumption. If we assume all latent variables share the same support size, higher-order tensors (e.g., three-way tensors) could also be used for identification. **This suggests that identifiability bounds may vary depending on the structural assumptions one makes.**
> > >
> > > Our theoretical results can be extended to hierarchical structures, enabling applications like those in [1]. Unlike [1], which requires an invertibility assumption for discrete component identification, our method only relies on the sufficient measurement assumption (implied by completeness). Moreover, we relax the pure-child assumption used in [1], allowing identification under sparse measurement settings. This broadens the applicability of our framework to more realistic scenarios.
> > >
> > > >[**Question 2**] I may consider increasing it further if a compelling real-world application scenario is provided.
> > >
> > > **A2**: Similar to [1], **our theoretical results can be useful in tasks like CLIP, by providing greater generality and explainability in modeling relationships between text and images**. Beyond this, the proposed method and the tensor rank condition are also applicable to biological data [2], gene expression studies [3], and social science domains such as psychology [4] — areas where latent confounders are prevalent and causal discovery under such conditions remains a significant open challenge. As an example, we demonstrate the applicability of our method on the Industrialization and Political Democracy dataset in Appendix H.1.
> > >
> > > ----------------
> > >
> > > Besides, we conducted another real-world experiment on the Big Five Personality dataset (https://openpsychometrics.org/) in psychology as [4]. The result is presented in https://anonymous.4open.science/r/Big5-6063/. It consists of nearly 20,000 data points. Here, we use the 10 corresponding indicators of Conscientiousness, Extraversion and Agreeableness to identify the latent factors and underlying causal structure. We chose the Chi-squared test to test independence among variables.
> > >
> > > **Acknowledgements.** Thank you for the constructive questions, which have inspired us to further explore the applicability of tensor rank and the broader identifiability results for latent variable models. We are glad that our responses addressed your concerns and we sincerely appreciate the improved evaluation. Please feel free to reach out if you have any further questions.
> > >
> > > **Reference**:
> > >
> > > [1] Learning Discrete Concepts in Latent Hierarchical Models. NeurIPS 2024.
> > >
> > > [2] Causal Representation Learning from Multimodal Biological Observations. ICLR 2025.
> > >
> > > [3] Automating the Selection of Proxy Variables of Unmeasured Confounders. ICML 2024.
> > >
> > > [4] A Versatile Causal Discovery Framework to Allow Causally-Related Hidden Variables. ICLR 2024.

---

### Official Review · Reviewer_e9yn · 2025-03-14

**Overall Recommendation:** 4

**Summary:**

When observational data contain both continuous and discrete variables, learning the causal structure among latent variables becomes a critical problem. Existing methods are often sensitive to parameter estimation. In this paper, the authors propose a statistically testable approach, the tensor rank condition, to address this issue. By discretizing continuous variables appropriately, the authors introduce graphical criteria for non-linear causal models with discrete latent variables, leveraging the tensor rank condition. Building on this, the authors formulate the Mixed LSMs framework and further develop a two-stage algorithm: first identifying causal clusters and then inferring the causal structure among latent variables. This approach provides a novel solution to the identification of Mixed LSMs. Experimental results demonstrate that the proposed algorithm outperforms state-of-the-art methods.

**Claims And Evidence:**

The claims are well supported by theoretical analysis and extensive experiments.

**Essential References Not Discussed:**

To the best of my knowledge, all key references are well-discussed in the related work section.

**Experimental Designs Or Analyses:**

Experimental designs and analyses are sound.

**Methods And Evaluation Criteria:**

The proposed method is reasonable and sound. The evaluation metrics chosen in this paper are appropriate for the focused problem, i.e., the identification of latent confounders.

**Other Comments Or Suggestions:**

NA.

**Other Strengths And Weaknesses:**

Strengths:

- The authors present graphical criteria for non-linear causal models with discrete latent confounders, establishing a connection between tensor rank and d-separation relations in the graph. This offers a novel methodology for studying causal structures in non-linear models.

- In Appendix D, the authors provide identifiability results for discrete latent structures under a sparsity condition, relaxing the pure-child requirement from previous work. This is an interesting contribution.

- The authors discuss the discretization process in detail, providing clear examples that make the methodology easier to follow.

Weaknesses:

- Estimating tensor rank in practice can be challenging, yet the paper does not provide sufficient discussion on this issue.

- The data generation process in the experiments follows a mixture model rather than directly following Eq. (1). The author should clarify this.

- The effectiveness of the discretization approach depends on accurately estimating the rank of the probability table. However, the paper does not discuss how the precision of rank estimation impacts causal structure learning.

**Questions For Authors:**

1. Previous works often assume sufficient observational support for latent variable identification. Why is the two-sufficient measurement condition enough for testing the tensor rank condition?

2. Why does the identifiability of Mixed LSMs require that at least one set of observed variables is caused by a single latent parent? If this is a structural assumption, it should be explicitly stated in the main text.

**Relation To Broader Scientific Literature:**

The proposed method is a novel solution for the latent confounder identification.  The proposed solution is both interesting and novel and is more realistic in practical scenarios.

**Theoretical Claims:**

The theoretical claims are well-discussed, and the proofs are presented clearly.

---

> ### Author Rebuttal · Authors · 2025-03-31
>
> We appreciate your valuable comments and suggestions and thank you for your positive assessment of our work.
>
> >[W1] Estimating tensor rank in practice can be challenging, yet the paper does not provide sufficient discussion on this issue.
>
> **W1A1**: As discussed in the Sec. 5 (Lines 402–404), we estimate matrix rank using the hypothesis test proposed by Mazaheri et al. (2023), and tensor rank following the approach in Chen et al. (2024). We will add more discussion on this point to improve clarity.
>
> >[W2] The data generation process in the experiments follows a mixture model rather than directly following Eq. (1). The author should clarify this.
>
> **W2A2**: Thank you for the comment. To ensure that the generated data satisfies the completeness condition, we use a mixture model as a practical and effective way to simulate data. Although the data is generated from a mixture model, the resulting joint distribution still reflects nonlinear dependencies (due to the probability property of the mixture model), and thus remains consistent with the setting described in Eq. (1). We will clarify this point in the revised text.
>
> >[W3] The effectiveness of the discretization approach depends on accurately estimating the rank of the probability table. However, the paper does not discuss how the precision of rank estimation impacts causal structure learning.
>
> **W3A3**: In our theoretical results, once the latent support size $r$ is identified, the causal structure can be recovered through rank-decomposition with specified $r$. Otherwise, an incorrectly estimated $r$ may lead to incorrect identification of latent structure, especially in the number of latent variables. To show that the latent support can be estimated consistently, we provide additional simulation results on estimating the rank of the probability table.
>
> We consider the following settings: $G_1$: $L_1 \to \{X_1, X_2, X_3, X_4\}$; $G_2$: $L_1 \to L_2$, $L_1 \to \{X_1, X_2, X_3\}$, $L_2 \to \{X_4, X_5, X_6\}$. In both settings, the latent variables have support size 2, and the observed variables have support size 3. Each experiment was repeated 1,000 times with randomly generated data. Using the hypothesis test by Mazaheri et al. (2023), we find that the rank of the probability table can be accurately estimated in most cases. This suggests that the effectiveness of our structure learning method, which relies on tensor rank, is supported by the reliability of rank estimation.
>
>
> ||Accuracy|Accuracy|Accuracy|Accuracy|
> |---|---|---|---|---|
> |#samples|3000|5000|10000|30000|
> |G1|0.82(±0.05)|0.85(±0.05)|0.84(±0.05)| 0.85(±0.06)|
> |G2|0.76(±0.06)|0.79(±0.06)|0.80(±0.06)| 0.82(±0.05)|
>
> >[Q1] Previous works often assume sufficient observational support for latent variable identification. Why is the two-sufficient measurement condition enough for testing the tensor rank condition?
>
>
> **Q1A1**: Thank you for the question. For each latent variable, two sufficient measurement variable $X_i, X_j$ are enough to identify the latent support by detecting the rank deficiency in the joint distribution $\mathbb{P}(X_i, X_j)$. Moreover, there is a key property of tensor rank: the rank of a tensor is at least as large as that of any of its subtensors,
> e.g., Rank$(\mathbb{P}(X_i, X_j, X_k))$ $\geq$ Rank$(\mathbb{P}(X_i, X_j, X_k=c))$.
>
> For example, consider an case that $L$ d-separated $X_i, X_j, X_k$, where $X_i, X_j$ are sufficient measurement variables and $L$ has the support size $r$. If the rank condition holds for a subtensor, e.g., Rank$(\mathbb{P}(X_i, X_j, X_k=c)) = r$, then the full tensor $\mathbb{P}(X_i, X_j, X_k)$ must also have rank $r$, regardless of whether $X_k$ satisfies the sufficient measurement condition (i.e., even if its support is smaller than $r$).
>
> Therefore, two sufficient measurements per latent variable are enough to test the tensor rank condition.
>
>
>
>
>
> >[Q2] Why does the identifiability of Mixed LSMs require that at least one set of observed variables is caused by a single latent parent? If this is a structural assumption, it should be explicitly stated in the main text.
>
>
> **Q2A2**: Thank you for pointing this out. This requirement is included in the Three-Pure-Child-Variable Assumption in Definition 4.1, and we further discuss it in Remark D.5. Broadly speaking, this assumption is used to identify the support of latent variables when considering the n-factor structure. We will make this assumption more explicit in the main text.

---

### Official Review · Reviewer_xpF6 · 2025-03-16

**Overall Recommendation:** 2

**Summary:**

This work proposes a causal discovery algorithm for some class of causal graph involving discrete latent variables and both discrete and continuous observed variables. The algorithm is essentially an extension of Chen et al. (2024) which uses rank tests on the probability tensors of observed variables to infer the graph connecting the latent variables. The novelty of the approach resides in its ability to deal with continuous observed variables by proposing a discretization scheme motivated by a theoretical analysis (Section 4.1).

**Claims And Evidence:**

I'm inclined to say that this work makes reasonable claims that are backed by sufficient evidence (both theoretical and empirical). That beings said, clarity is an important issue which prevents me from recommending acceptation of this paper. I found many cases where theoretical claims are so unclear that it is very difficult to assess whether they are sound or not. More on this later.

**Essential References Not Discussed:**

I didn't notice anything obvious.

**Experimental Designs Or Analyses:**

See above.

**Methods And Evaluation Criteria:**

The paper is mainly theoretical, but does present some empirical validation on synthetic data. This sort of empirical analysis is fairly standard for a theoretical work and IMO sufficient. That being said, I highly recommend the authors to find either a more realistic dataset to test their approach or at least a motivating example. Right now the approach feels a bit unmotivated.

**Other Comments Or Suggestions:**

Line 429: Typo

**Other Strengths And Weaknesses:**

Strengths:

- I thought the topic was interesting, the title and the general theme got me curious and stimulated. I believe this is overall an interesting direction.
- I appreciate the effort made by the authors to include a running example, which helped grounding some of the technical concepts.

Weakness:

- The work felt a bit unmotivated. IIRC, next to no effort is made to explain why this direction is exciting and could lead to interesting applications in the future.
- Novelty is between low and moderate. If I understood correctly, the only algorithmic novelty comes from the discretization phase which then allows the application of Chen et al. (2024), which works for discrete variables. Section 4.1 also contains a theoretical analysis of discretization. This feels a bit marginal, but I suppose this is subjective.
- The most important issue with this work is its lack of clarity. Below, I give a list of confusions I got as I read the manuscript, which I believe are due to poor writing.

Clarity:
- “In the causal graph, we do not allow directions from observed to latent to preserve the latent confounder structure.” Do you allow edges from observed to observed variables? It seems all the figures have graphs without such edges, but you do not mention it anywhere AFAIK. Maybe these assumptions should belong to an Assumption environment given their importance? Oh I see this assumption is done much later in Definition 4.1. Overall it seems like multiple results use different assumptions. It would be useful to refer to these definitions/assumptions using \ref directly in the theorems, otherwise it’s hard to follow which assumptions are made and where.
- Eq (1), f should be indexed by i, no?
- Assumption 2.1 (a): Do you mean for all $r \in \Omega$ here? As in, each marginal P(L_i) puts mass everywhere on $\Omega$? If so, this is not clear here.
- Assumption 2.1 (b): Might be useful to say explicitly what this conditional distribution contingency table is. What is its shape? Which dimension corresponds to which variable?
- Theorem 3.2:
    - It would be nice to have a definition of what is meant by the rank of a tensor. I would argue that this is not a broadly known notion (unlike the rank of a matrix for example)
    - X_p = {X_1, …, X_n} but then it’s written X_p \subseteq X… But earlier it was said that X was n-dimensional. So X_p is just the set of observed variables here? Or is it a subset?
    - Can the conditional set S intersect with X_p? X? Or should it be constrained to the latent variables?
- General point: Looks like there’s a clash in notation. Lower case r is used both for the rank of tensor and for the cardinality of \Omega.
- Condition 3.3:
    - I’m a bit confused here. The condition concerns only the variables X_i that satisfy the condition “for all L_j, X_i \indep X\X_i | L_j”, correct? I couldn’t find a graph in your figures that had such a node X_i. Could you give an example of such a variable in a given graph?
    - Isn’t it the case that any function $g: \Omega \rightarrow \mathbb{R}$ is bounded since $\Omega$ is finite? Actually, are we assuming it’s finite? Also, am I right to assume that the codomain of $g$ is the real line?
    - Are we saying that completeness holds for all conditionals P(L_j | X_i)? I.e. for all pairs (X_i, L_j)? That’s unclear because there’s no quantifier for i and j here.
    - It seems completeness is a property of the joint P(L_j, X_i), not just the conditional P(L_j | X_i) since  statements like E[g(L_j) | X_i] = 0 almost surely depends on the distribution P(X_i). Same for g(L_j) = 0 a.e., which cannot be verified from P(L_j | X_i) alone (you need the marginal over L_j).
    - Appendix F on condition 3.3 didn’t help and is poorly written, I really didn’t get the point there.
- Theorem 3.4
    - Same issues as in Theorem 3.2
    - It seems we are manipulating a Tensor with n-dimensions where some of its dimensions have a continuum of indices. How do you define rank here? Also, this is a very unusual object so it should be described and explained more.
- Definition 4.1
    - what’s a pure variable? It’s not defined here.
    - what’s a “sufficient measured variable”?
- Line 244: The “tensor rank condition” is constantly mentioned, but is not properly defined anywhere. I genuinely don’t know what is meant here. I think what is actually meant here is whether we can find a discretization that will yield a tensor that satisfies the assumptions of Theorem 3.2, right?
- Proof of Proposition 4.2:
    - Can you point to where we are assuming that each observed variable has a single latent parent? (you use this assumption on line ~266).
    - Does this proposition mean that any discretization will work? (as long as it’s not clustering all values in a single state?
- Theorem 4.5: What’s k here? The iteration of some algorithm? Which algorithm?
- Definition 4.9: Typo here? n should be p no?
- Proposition 4.15: Isn’t it weird that X_s does not appear in (i), given the quantifier “for all X_s”?

**Questions For Authors:**

My questions appears above, and concern mainly the clarity of the work.

**Relation To Broader Scientific Literature:**

I am not following this literature closely, but it appeared to me that the contribution is properly contextualized within existing works. My understanding is that this work is an extension of Chen et al. (2024), which is made very clear and transparent in the manuscript, to deal with continuous observed variables. That being said, the delta between both works appears to be fairly small (it adds discretization strategy backed by theoretical analysis). Maybe the authors could explain a bit more how their work differs from Chen's?

**Theoretical Claims:**

I did read all the theoretical claims made in the main paper, but did not read the proofs in the appendix. The results appears believable, but clarity is a very significant issue IMO. (more on this later)

---

> ### Author Rebuttal · Authors · 2025-03-31
>
> Thank you for your careful review. We address each point below and have corrected the typos. Please feel free to reach out with any further questions (due to limited space).
>
> >W1W2: ... realistic dataset to test their approach.
>
> A: Please see Appendix H.1.
> >W3: Novelty is between low and moderate.
>
> A: Please see the **Q1A1** response of #Review gVAe.
> >Q1: Do you allow edges from observed to observed variables?
>
> A: We allow edges between observed variables. This highlights that the tensor rank condition applies beyond the pure measurement setting (e.g., impure case in Appendix D).
> >Q3: Assp. 2.1 (a): Do you mean for all $r\in \Omega$ here?
>
> A:  Yes, we mean that for any $r \in \Omega$ here — each marginal places non-zero mass on every element of $\Omega$.
> >Q4: Assp. 2.1 (b):  What is its shape? Which dimension corresponds to which variable?
>
> A: The conditional distribution can be represented as a contingency table— a $|V_i| \times |Pa(V_i)|$ matrix.
> >Q5.1: Theo. 3.2: ...what is meant by the rank of a tensor.
>
> A: The rank of a tensor $\mathcal{X}$, denoted $rank(\mathcal{X})$, is the smallest number of rank-one tensors that generate $\mathcal{X}$ as their sum, where a $N$-way tensor is rank-one if it can be written as the outer product of $N$ vectors.
> >Q5.2: Theo. 3.2: So $X_p$ is just the set of observed variables here? Or is it a subset?
>
> A: We modify it to $\mathbf{X}_p =\{X_1,\cdots,X_{p}\} \subseteq \mathbf{X}, p \leq n$.
> >Q5.3: Theo. 3.2: Can the conditional set $S$ intersect with $X_p$? Or should it be constrained to the latent variables?
>
> A: Yes, we allow $S \cap X_p \neq \emptyset$ and allow observed variables included in $S$.
> >Q6: Lower case r is used both for the rank of tensor and for the cardinality of $\Omega$.
>
> A: We will use distinct symbols for tensor rank and the cardinality of $\Omega$ in revision.
> >Q7.1: Cond. 3.3: ...only X_i that satisfy “for all L_j, $ X_i \bot  X\setminus X_i | L_j$”, correct?... Could you give an example? Q7.2: ...function $g:\Omega \to R$ is bounded since $\Omega$ is finite? Q7.3: ...no quantifier for $i$ and $j$ here.
>
> A: We will clarify this in the revision: for any $X_i \in \mathbf{X}$ that has only one (latent) parent $L_j$, we assume the conditional distribution $\mathbb{P}(L_j|X_i)$ is complete. That is, for all measurable real function $g$ such that $\mathbb{E}(|g(l)|) < +\infty$， $\mathbb{E}(g(l)|x) = 0$ almost surely iif $g(l) = 0$ almost surely. As an example, consider $X_7, L_3$ in Fig. 2. Besides, we assume that the domain $\Omega$ is finite. In this case, any function $g:\Omega \to R$ is indeed bounded.
> >Q7.4: Cond. 3.3: It seems completeness is a property of the joint $P(L_j, X_i)$...Q7.5: App. F on cond. 3.3 didn’t help.
>
> A: Yes, completeness is the property of the joint distribution. Due to the Bayesian formula, the completeness on the conditional distribution $P(L_j|X_i)=P(L_j, X_i)/ P(X_i)$ involves the joint distribution. In this paper, we adopt the conditional form for consistency with prior work (e.g., Cui et al., 2023).
> In Appendix F, we illustrate that the completeness condition is not overly restrictive and can arise naturally in practice. We’ve revised the appendix to clarify this intention.
> >Q8: Theo. 3.4: How do you define rank here?
>
> A: As noted in Remark G.3, we treat the tensor as a theoretical representation of the joint distribution, with rank defined by its minimal rank-one decomposition. We will clarify this in the revision.
> >Q9: Def. 4.1: What’s a pure variable? It’s not defined here. What’s a “sufficient measured variable”?
>
> A:  Pure variables denote the variables that have only one latent parent, and no observed parents. For an observed variable $X$ with support $\mathcal{X}$ and latent parent $L$ with support $\Omega$, we define a sufficient measurement as $|\mathcal{X}| > |\Omega|$.
> >Q10: Line 244: The “tensor rank condition” ... is not properly defined anywhere...
>
> A: The rank condition refer the rank deficiency of probability tensor. Besides, your statement is right, we would like to study when and how to use discretization to make this properity hold in continous data.
> >Q11.1: where we are assuming that each observed variable has a single latent parent?
>
> A: This assumption follows from the model definition, where we adopt the standard pure measurement setting (Silva et al., 2006), in which each observed variable has a single latent parent. We will clarify this more explicitly in the revision.
> >Q11.2: Does this proposition mean that any discretization will work?
>
> A: No, not all discretizations will work. We clarify this point in Remark 4.3 and illustrate it with an example in Appendix B.
> >Q12: Theo. 4.5: What’s k here? The iteration of some algorithm?
>
> A: k refers to the number of discretization steps, as discussed in Remark 4.6.
> >Q14: Prop. 4.15: Isn’t it weird that $X_s$ does not appear in (i)...?
>
> A:  $X_s$ does not need to appear in (i) since it may not share a common latent parent with the other variables.

---

### Decision · Program_Chairs · 2025-05-01

**Decision:**

Accept (poster)

**Comment:**

While two reviewers knowledgeable of the field rated this work highly, one reviewer raised concerns regarding the clarity of the work. While authors addressed many concerns during the rebuttal period, this remains a paper that only only a small fraction of the ICML readership may be able to follow. I therefore recommend a weak accept.